# CRELD1 is an evolutionarily-conserved maturational enhancer of ionotropic acetylcholine receptors

**Manuela D'Alessandro**[1], **Magali Richard**[1†], **Christian Stigloher**[1‡], **Vincent Gache**[1], **Thomas Boulin**[1], **Janet E Richmond**[2], **Jean-Louis Bessereau**[1*]

[1]Univ Lyon, Université Claude Bernard Lyon 1, CNRS UMR 5310, INSERM U 1217, Institut NeuroMyoGène, Lyon, France; [2]Department of Biological Sciences, University of Illinois at Chicago, Chicago, United States

**Abstract** The assembly of neurotransmitter receptors in the endoplasmic reticulum limits the number of receptors delivered to the plasma membrane, ultimately controlling neurotransmitter sensitivity and synaptic transfer function. In a forward genetic screen conducted in the nematode *C. elegans*, we identified *crld-1* as a gene required for the synaptic expression of ionotropic acetylcholine receptors (AChR). We demonstrated that the CRLD-1A isoform is a membrane-associated ER-resident protein disulfide isomerase (PDI). It physically interacts with AChRs and promotes the assembly of AChR subunits in the ER. Mutations of *Creld1,* the human ortholog of *crld-1a,* are responsible for developmental cardiac defects. We showed that *Creld1* knockdown in mouse muscle cells decreased surface expression of AChRs and that expression of mouse *Creld1* in *C. elegans* rescued *crld-1a* mutant phenotypes. Altogether these results identify a novel and evolutionarily-conserved maturational enhancer of AChR biogenesis, which controls the abundance of functional receptors at the cell surface.
DOI: https://doi.org/10.7554/eLife.39649.001

**\*For correspondence:**
jean-louis.bessereau@univ-lyon1.fr

**Present address:** †Laboratoire TIMC-IMAG, UMR 5525, CNRS, Université Grenoble Alpes, Grenoble, France; ‡Imaging Core Facility, Biocenter of the University of Würzburg, Würzburg, Germany

**Competing interests:** The authors declare that no competing interests exist.

## Introduction

The total amount or neurotransmitter receptors synthesized within a neuron or a muscle cell determines the size of the receptor pool that can be delivered to the plasma membrane and, ultimately, the responsiveness of the cell to specific transmitters. Assembly of multiple subunits into mature receptors in the ER seems to be an inefficient and limiting step in the synthesis of ligand-gated ion channels (LGCI) belonging to the Cys-loop superfamily of receptors (*Crespi et al., 2018*; *Fu et al., 2016*; *Herguedas et al., 2013*; *Jacob et al., 2008*).

This family, which was initially defined based on nicotinic acetylcholine receptors (AChRs), also includes GABA$_A$, glycine and serotonin 5-HT3 receptors. They are made of five identical or homologous subunits arranged around a fivefold pseudo-symmetrical axis (*Albuquerque et al., 2009*; *Cecchini and Changeux, 2015*; *Du et al., 2015*; *Hassaine et al., 2014*; *Miller and Aricescu, 2014*; *Morales-Perez et al., 2016*). Each subunit has a large amino-terminal extracellular region, a trans-membrane domain containing four alpha-helical segments (M1-M4), and a variable hydrophilic cytoplasmic loop between M3 and M4. Extracellular regions tightly interact to form a doughnut-like structure containing the ligand binding sites. Upon ligand binding, receptor rearrangements cause the opening of a central ion channel lined by the M2 segments contributed by each of the five subunits. The assembly of such large pentameric complexes (250–300 kDa), each containing twenty trans-membrane domains, is challenging for the cellular machinery. For example, the half-life of the AChR assembly is 90 min whereas the influenza hemagglutinin takes only 7–10 min to form homotrimers (*Wanamaker et al., 2003*). In muscle cells, only 30% of the synthesized alpha-subunits of AChRs are

assembled into heteromeric receptors and in neurons, a pool of immature receptors is readily detectable in intracellular compartments (*Arroyo-Jim nez et al., 1999*; *Henderson and Lester, 2015*). Similarly, only 25% of synthesized GABA$_A$ receptors are assembled into mature receptors (*Gorrie et al., 1997*; *Wanamaker et al., 2003*).

Early work on AChRs in muscle cells showed that pentamerization was sequential and intermediate complexes containing only two or three subunits were identified (*Colombo et al., 2013*). Interactions with general chaperone proteins and components of the ER biosynthetic machinery, such as BiP (*Blount and Merlie, 1991*), calnexin (*Gelman et al., 1995*; *Wanamaker and Green, 2005*) and ERp57/Protein disulfide-isomerase A3 (*Wanamaker and Green, 2007*), assist correct folding and protect immature intermediates from ER-associated degradation (ERAD). Other proteins such as RIC3, NACHO, and members of the LY6 prototoxin family more specifically control the assembly or the stoichiometry of particular AChR subtypes. Interestingly, modulating AChR assembly in the ER has important impacts on cell homeostasis. Nicotine, the main tobacco component responsible for smoking addiction (*Picciotto and Mineur, 2014*; *Subramaniyan and Dani, 2015*), was demonstrated to bind early AChR assembly intermediates in the ER and promote receptor maturation by acting as a pharmacological chaperone. This effect partly accounts for the upregulation of AChR expression caused by chronic exposure to nicotine in the brain (*Sallette et al., 2005*; *Henderson and Lester, 2015*). Nicotine was also recently shown to protect dopaminergic neurons from mild ER stress partly by promoting the export of AChRs and consequently diminishing the unfolded protein response (UPR) (*Srinivasan et al., 2016*). In contrast, mutations impairing the export of $\alpha_4\beta_2$ AChRs from the ER were associated with amyotrophic lateral sclerosis (*Richards et al., 2011*).

Despite the pathological and physiological importance of AChR biogenesis at early steps, this process still remains poorly characterized. Genetic screens in model organisms represent a valid tool to further elucidate this biosynthetic pathway because they are able to identify relevant factors irrespective of their abundance or the stability of their biochemical interactions with other components of the pathway. Here we used *C. elegans* as a model organism to dissect the genes involved in AChR biosynthesis. ACh is the main excitatory neurotransmitter in *C. elegans* and at least 30 genes encode AChR subunits (*Holden-Dye et al., 2013*). Two types of ionotropic AChRs, heteromeric levamisole-sensitive AChRs (L-AChRs) and homomeric nicotine-sensitive AChRs (N-AChRs), are present at neuromuscular junctions (*Richmond and Jorgensen, 1999*). L-AChRs can be activated by the nematode-specific cholinergic agonist levamisole, which causes hypercontraction of *C. elegans* body-wall muscles (BWMs) and death of wild-type worms at high concentrations (*Fleming et al., 1997*; *Lewis et al., 1980*). Genetic screens for complete resistance to levamisole have identified the structural subunits of the receptor, including three α subunits (LEV-8, UNC-38, and UNC-63) and two non-α subunits (LEV-1 and UNC-29). The second type of receptor, N-AChR, is activated by nicotine and is insensitive to levamisole.

To identify factors involved in the biogenesis of L-AChRs, we performed a genetic screen for mutants with decreased sensitivity to levamisole and identified *crld-1*, the ortholog of the vertebrate genes *Creld1* and *Creld2* (Cystein-Rich with EGF-Like Domains). *Creld1* is widely expressed in humans, with the highest levels in adult heart, brain and skeletal muscle (*Rupp et al., 2002*). Missense mutations in human CRELD1 are linked to atrioventricular septal defects (AVSD) (*Robinson et al., 2003*) and in mice *Creld1*$^{-/-}$ embryos die at embryonic day 11.5 with several defects including heart development defects (*Mass et al., 2014*). Allelic interactions between CRELD1 and VEGFA (vascular endothelial growth factor-A) contribute to AVSD (*Redig et al., 2014*) and it was proposed that CRELD1 is required for VEGF-dependent proliferation of endocardial cells by promoting expression of NFATc1 target genes (*Mass et al., 2014*). CRELD2 was identified as an ER stress-inducible gene. The protein seems to predominantly localize to the ER and Golgi apparatus, but some reports suggested that it might also be secreted when overexpressed (*Oh-hashi et al., 2011*). Interestingly, the function of CRELD proteins at the neuromuscular junction has not been previously investigated.

Here we demonstrate that *crld-1* is required for AChR biogenesis and behaves as a maturational enhancer by promoting the assembly of L-AChR subunits in the ER.

## Results

### Disruption of the evolutionarily conserved gene *crld-1* confers partial resistance to the cholinergic agonist levamisole

To identify genes regulating the activity of L-AChRs in *C. elegans*, we used *Mos1*-mediated insertional mutagenesis (*Boulin and Bessereau, 2007*; *Williams et al., 2005*) and screened for mutants with only partially decreased sensitivity to levamisole, because screens for complete resistance are likely saturated. Such mutants completely paralyze on high levamisole concentrations within a few hours but subsequently adapt within 12–16 hr and recover motility in contrast to the wild type (*Gally et al., 2004*; *Lewis et al., 1980*). We isolated two independent strains containing a *Mos1* insertion in the *F09E8.2* locus (*Figure 1A,B*), which we tentatively named *crld-1* because it is the sole *C. elegans* gene encoding proteins of the CRELD family (*Rupp et al., 2002*). *Crld-1* generates two transcripts, *crld-1a* and *crld-1b*, by alternative splicing of the last exons. Surprisingly, resistance to levamisole of the *kr133* mutants, that contain a *Mos1* insertion in the fourth exon shared by both transcripts, was less pronounced than in *kr132* mutants, which contain a *Mos1* insertion in the last *crld-1a*-specific exon. RT-PCR analysis of the *crld-1(kr133)* transcripts revealed that cryptic splice donor sites present in the *Mos1 kr133* transposon were used at low frequency to generate in-frame mRNAs (*Figure 1—figure supplement 1*). Therefore, *kr133* is likely to be a hypomorphic mutation.

To fully inactivate *crld-1*, we used the *tm3993* allele, which contains a deletion of the first three exons, and we also engineered a null allele by inserting a 2.8 kb HySOG dual selection cassette in the first *crld-1* coding exon (*kr297*). None of the *crld-1* mutants exhibited a grossly abnormal phenotype. Specifically, locomotion remained coordinated and only a slight decrease of the thrashing frequency in liquid could be detected in *tm3993* mutants but not in *Mos1* alleles. The most dramatic phenotype was a decreased sensitivity to levamisole since almost 100% of the *crld-1* mutant animals fully adapted overnight to 1 mM levamisole while all the wild-type animals were paralyzed (*Figure 1C*).

The *C. elegans* CRLD-1A and -1B are predicted to be 356 and 310 amino acid proteins, respectively, containing a signal peptide, a N-terminal region rich in glutamic acid and tryptophan residues called a DUF3456 or WE domain (*Finn et al., 2016*; *Mass et al., 2014*; *Rupp et al., 2002*) and 3 EGF-like domains (*Figure 1B*). CRLD-1 isoforms differ at their C-terminus: CRLD-1A ends with two predicted transmembrane domains whereas CRLD-1B ends with a KDEL sequence, which is an endoplasmic reticulum (ER) retention signal (*Figure 1B*). This modular organization was highly conserved among CRELD proteins throughout evolution (*Rupp et al., 2002*). Interestingly in vertebrates such as fish, mouse and human, the transmembrane and the non-transmembrane CRELD proteins are encoded by two distinct genes, *Creld1* and *Creld2* (*Maslen et al., 2006*; *Rupp et al., 2002*), respectively.

Since the *kr132* mutation was predicted to only impair the *crld-1a* transcript, we tested whether the transmembrane CRLD-1A was specifically required for normal levamisole sensitivity or whether both isoforms were necessary. First, we knocked-in the GFP-coding sequence just after the predicted signal peptide in the *crld-1* locus. The resulting allele *crld-1(kr298::gfp)* had wild-type sensitivity to levamisole and provided a means to visualize the expression pattern of both CRLD-1 isoforms (*Figure 1C*). Second, we used the Co-CRISPR technique (*Arribere et al., 2014*) to generate isoform specific mutants. We replaced the splicing acceptor site of exon −9a and −9b of *crld-1(kr298::gfp)* with a stop codon in order to suppress the expression of *crld-1a* and *crld-1b* isoforms, respectively (*Figure 1A*). Interestingly, *crld-1a*-specific mutants were as resistant to levamisole as *crld-1(tm3993)* while *crld-1b*-specific mutants were indistinguishable from the wild type (*Figure 1C*). To confirm that CRLD-1A was necessary and sufficient for levamisole sensitivity, we independently expressed either *crld-1a* or *crld-1b* cDNAs in body-wall muscle and observed that the A isoform could rescue the levamisole sensitivity of *tm3993* null mutants, while the B isoform could not (*Figure 1D*).

Altogether, these data demonstrate that the transmembrane isoform CRLD-1A was the only isoform required for levamisole sensitivity and might act cell-autonomously to regulate L-AChR functional expression in body-wall muscle.

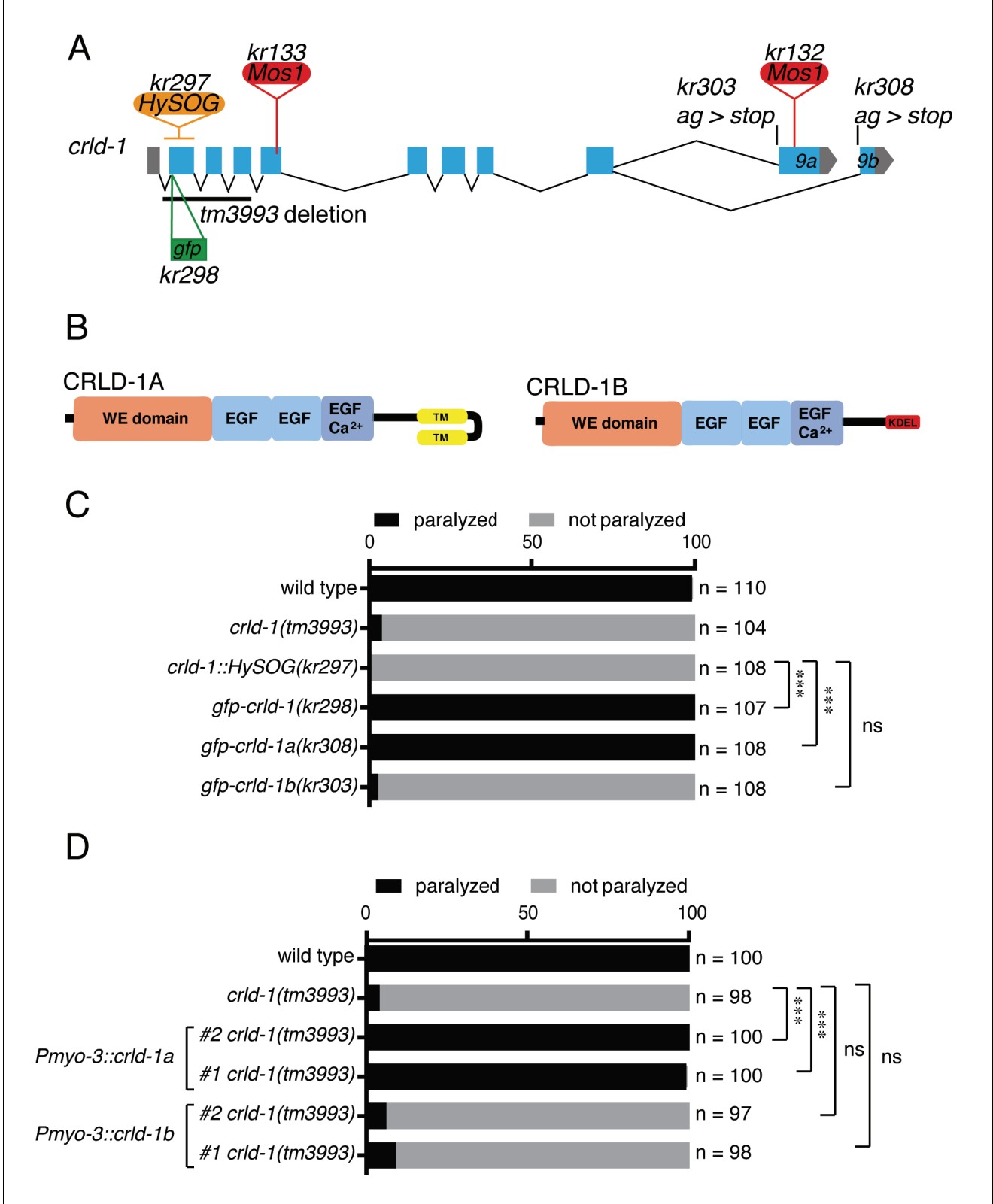

**Figure 1.** CRLD-1A isoform is sufficient for L-AChR expression based on sensitivity to levamisole. (**A**) Structure of the *crld-1* locus, which generates two isoforms (*crld-1a* and *crld-1b*) by alternative splicing of the last exon (exon *9a* and exon *9b*). The different *Mos1* transposon insertions and the mutant alleles are indicated. *kr303* and *kr308* mutations specifically express only *crld-1b* and *crld-1a*, respectively. HySOG = hygromycinB miniSOG dual selection cassette (length = 2.8 kb). The green box indicates the position of the GFP sequence inserted in the first exon of *crld-1* to generate the *gfp-*

*Figure 1 continued*

crld-1(kr298) knock-in allele. (**B**) Domain organization of CRLD-1A and CRLD-1B. SP = signal peptide, WE domain = tryptophan (**W**) and glutamic acid (**E**) enriched domain, EGF = Epidermal Growth Factor-like domain, EGF Ca$^{2+}$ = Ca$^{2+}$binding epidermal growth factor-like domain, TM = transmembrane domain, KDEL = Lys-Asp-Glu-Leu ER retention signal. (**C**) *crld-1* is necessary for wild-type sensitivity to levamisole. Gray bars indicate the percentage of moving animals after overnight exposure to 1 mM levamisole, and black bars indicate the percentage of paralyzed animals. Experiments were repeated three times, n = number of animals tested. p=0,2465, ns = not significant, \*\*\*p<0.001, after Bonferroni correction, Fisher exact probability test. (**D**) Body wall muscle expression of *crld-1a* but not *crld-1b* rescues levamisole sensitivity in *creld-1(tm3993)* mutants. Gray bars indicate the percentage of moving animals after overnight exposure to 1 mM levamisole, and black bars indicate the percentage of paralyzed animals. Two independent transgenic lines were tested for each condition. Experiments were repeated four times, n = number of animals tested. p=0,2504 and 0,5369, ns = not significant, \*\*\*p<0.001, after Bonferroni correction, Fisher exact probability test.

DOI: https://doi.org/10.7554/eLife.39649.002

The following figure supplement is available for figure 1:

**Figure supplement 1.** - Characterization of the *kr133* mutant allele.

DOI: https://doi.org/10.7554/eLife.39649.003

## CRLD-1A and CRLD-1B are ubiquitously expressed and localize to the ER

Analysis of *crld-1* expression using *gfp-crld-1* knock-in strains indicated that CRLD-1 was a ubiquitous protein (*Figure 2A–C*), as suggested initially by the expression of GFP from a *crld-1* transcriptional reporter in a multicopy transgene (*Figure 2—figure supplement 1A*). CRLD-1A and CRLD-1B were expressed in most, if not all cells including body-wall muscles, neurons, pharynx, hypodermis, seam cells, intestine and gonad (*Figure 2A–C*). In every cell type GFP-CRLD-1 had a reticular pattern.

We focused our analysis on body-wall muscles and found that the transmembrane CRLD-1A isoform localized to a perinuclear network highly reminiscent of ER localization. CRLD-1B distribution was similar, yet the perinuclear localization was less intense and a punctate pattern was superimposed onto the network distribution (*Figure 2A*). As expected *crld-1(kr298::gfp)* animals expressing both *crld-1a* and *crld-1b* isoforms tagged with *gfp* displayed a combination of the two patterns (*Figure 2—figure supplement 1B*).

To confirm that CRLD-1A and −1B did localize in the ER, we expressed a TagRFP-T fused to the ER retention signal KDEL in the muscle cells of *gfp-crld-1* knock-in animals. Both isoforms co-localized with the TagRFP-T-KDEL (*Figure 2D*). Interestingly, the TagRFP-T-KDEL had a punctate distribution very similar to CRELD-1B, suggesting that CRELD-1B is indeed retrieved to the ER by a KDEL-dependent mechanism. Conversely the Golgi marker α-MannosidaseII-TagRFP-T did not colocalize with either CRLD-1 isoforms (*Figure 2E*). These data demonstrated that both CRLD-1 isoforms primarily localized to the ER. The fact that CRLD-1A is likely to be in the ER membrane and CRLD-1B is likely in the ER lumen might account for the partially different distribution of the two isoforms within the muscle ER.

## CRLD-1 is required for cell surface expression of L-AChRs

The resistance of *crld-1* mutants to levamisole suggested that *crld-1* was required for the proper expression of functional L-AChRs at NMJs. To test this hypothesis, we used immunofluorescence to characterize cholinergic NMJs (*Figure 3A*). The number of cholinergic boutons, stained by the vesicular acetylcholine transporter UNC-17, was similar in the wild type and in *crld-1(tm3993)* mutants. In contrast, we observed an obvious decrease of synaptic L-AChRs stained by anti-UNC-38 antibodies in *crld-1(tm3993)* as compared to the wild type (*Figure 3A*). To quantify L-AChR content at NMJs we used a knock-in strain in which the red fluorescent protein TagRFP is fused to the essential L-AChR subunit UNC-29 (*Richard et al., 2013*). We found that the fluorescence intensity of L-AChRs present at the ventral nerve cord was decreased by 85% in *crld-1* mutant as compared to the wild type (*Figure 3F,G*). This could be due to reduced expression of L-AChRs by the muscle cells or to a redistribution of the receptors outside of the synapse. To discriminate between these hypotheses, we recorded the electrophysiological response of body-wall muscle to pressure-ejected levamisole and found a 65% decrease in the response of *crld-1(tm3993)* mutants compared to the wild type (*Figure 3B*). These results suggested that disrupting *crld-1* impairs surface expression of functional L-AChRs.

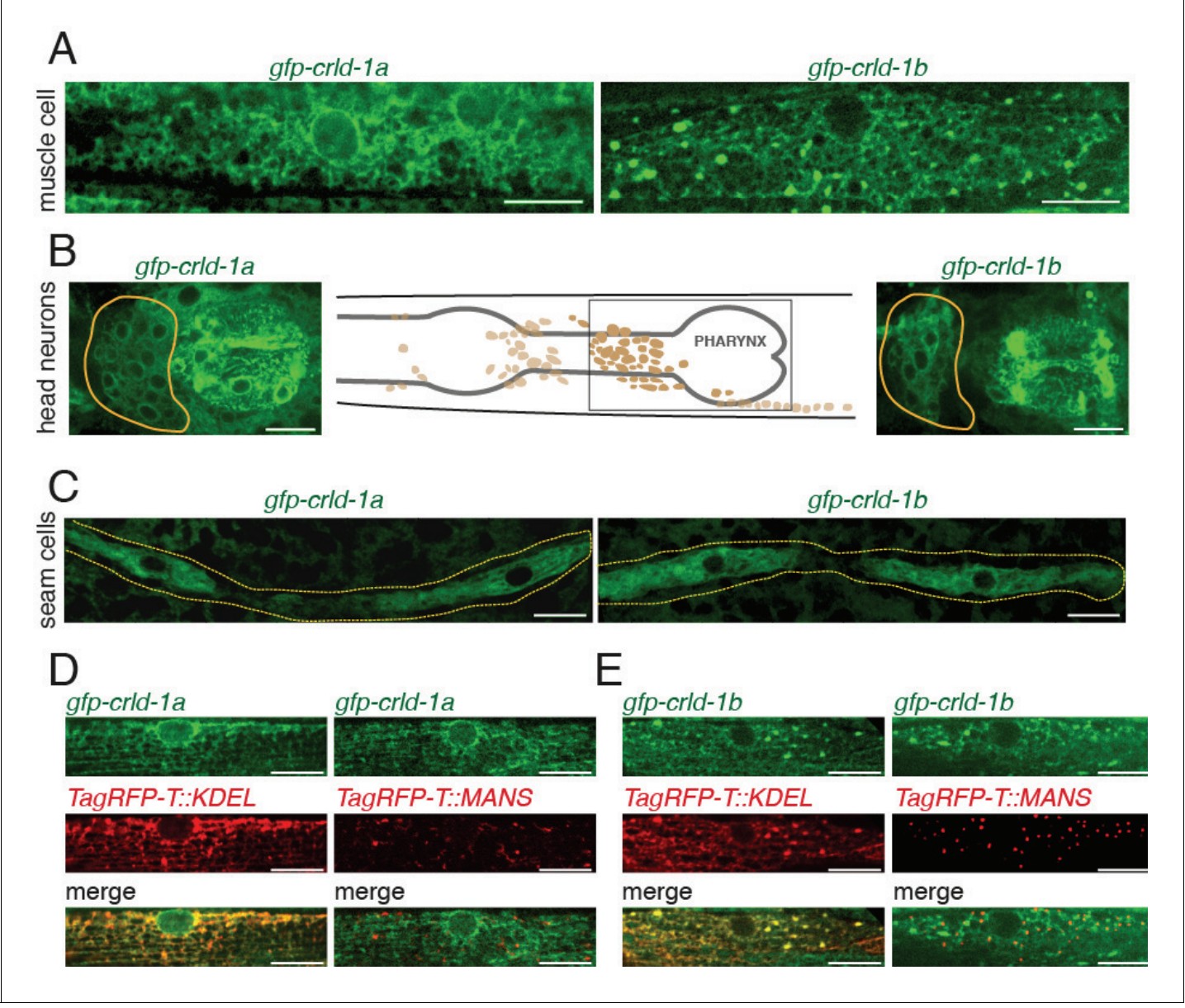

**Figure 2.** CRLD-1 is ubiquitously expressed and localizes in the ER of BWMs. (A) Distribution of GFP-CRLD-1A (left) and GFP-CRLD-1B (right) in muscle cells of *gfp-crld-1* isoform-specific knock-in worms. (B) Localization of CRLD-1A (left) and CRLD-1B (right) in the pharynx and in the lateral ganglion (encircled in yellow). The middle panel shows a schematic representation of the locations of neurons and ganglia in the head, adapted from: http://www.wormatlas.org/ver1/MoW_built0.92/nervous_system.html. (C) Localization of CRLD-1A (left) and CRLD-1B (right) in the epithelial seam cells of *gfp-crld-1* isoform-specific knock-in worms. Dashed lines, seam cell outlines. (D) Expression of the ER marker TagRFP-T::KDEL in *gfp-crld-1a* (left) and *gfp-crld-1b* (right) isoform-specific knock-in strains. TagRFP-T::KDEL displays a reticular pattern throughout the cytoplasm surrounding the nucleus that co-localizes with both CRLD-1A and CRLD-1B signals. (E) CRLD-1A and CRLD-1B from *gfp-crld-1a* (left) and *gfp-crld-1b* (right) knock-in animals do not co-localize with a Golgi-resident TagRFP-T-tagged Mannosidase II protein (MANS::TagRFP-T). In (D) and (E), the *Pmyo-3* promoter was used for expression of both TagRFP-T::KDEL and MANS::TagRFP-T in body wall muscles. In all panels, scale bars equal 10 μm.

DOI: https://doi.org/10.7554/eLife.39649.004

The following figure supplement is available for figure 2:

**Figure supplement 1.** - CRLD-1 expression pattern.

DOI: https://doi.org/10.7554/eLife.39649.005

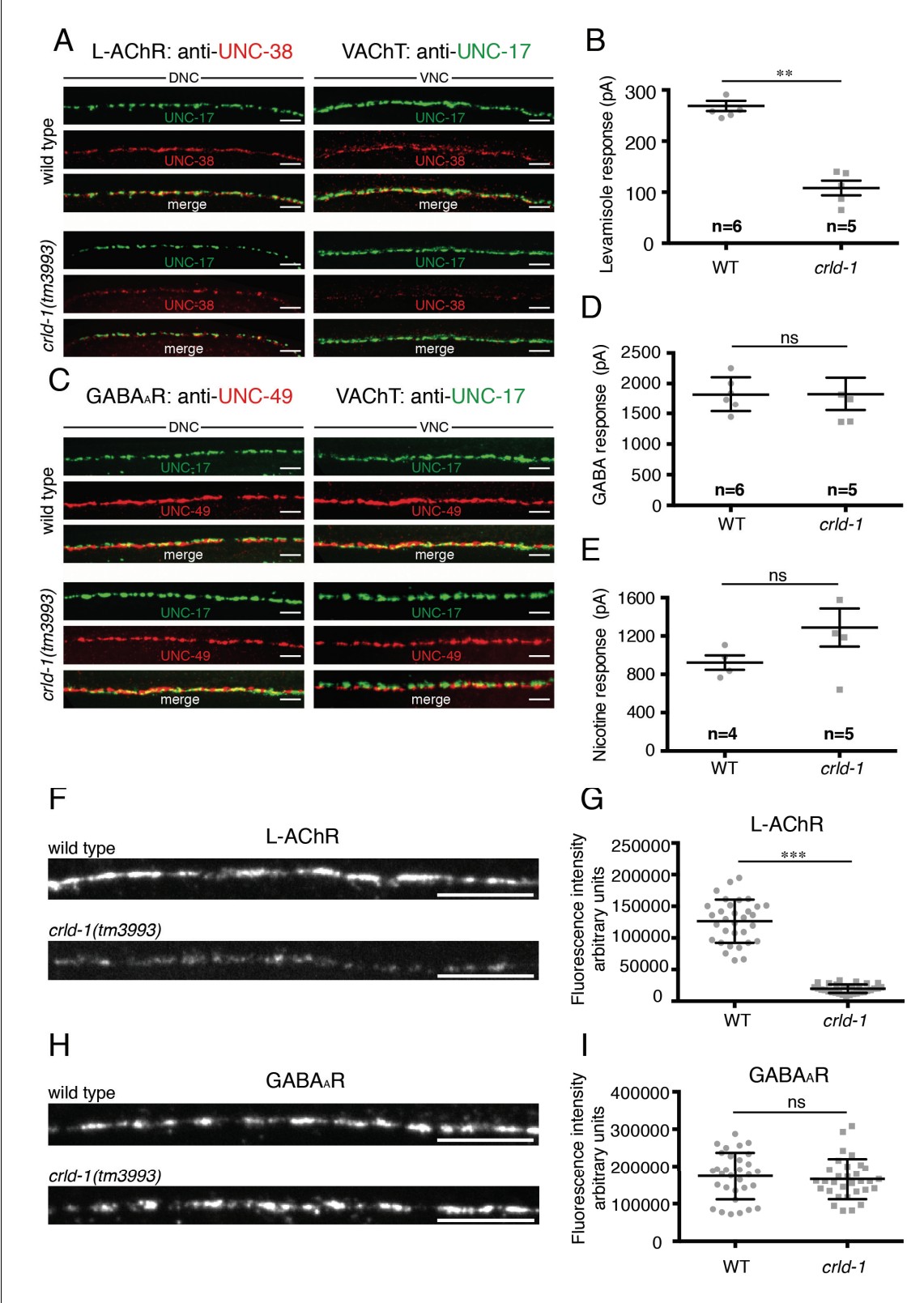

**Figure 3.** CRLD-1 is required for surface expression of L-AChRs. (**A**) L-AChR expression is decreased at NMJs of *crld-1(tm3993)* mutants, whereas presynaptic differentiation is unaffected. L-AChRs are labeled using anti-UNC-38. Cholinergic boutons are labeled using an anti-vesicular acetylcholine transporter UNC-17 (VAChT) antibody. DNC = dorsal nerve cord, VNC = ventral nerve cord. Scale bars 10 μm. (**B**) Response to pressure-ejection of levamisole in voltage-clamped ventral BWMs is reduced in *crld-1(tm3993)*. Data indicate mean ± SEM; WT: 269 ± 10 pA, n = 6 animals; *crld-1(tm3993)*: *Figure 3 continued on next page*

Figure 3 continued

108 ± 14 pA, n = 5 animals; p=0.0043. Mann-Whitney test. (C) GABA$_A$R expression is unaffected at NMJs of *crld-1(tm3993)* mutants compared to wild type. GABA$_A$R are labeled using anti-UNC-49 antibodies. Cholinergic boutons are labeled using anti-UNC-17 (VAChT) antibodies. DNC = dorsal nerve cord, VNC = ventral nerve cord. Scale bars 10 μm. (D) Electrophysiological response of body-wall muscle cells to pressure-ejection of GABA in *crld-1 (tm3993)* mutant is similar to the wild type. Data indicate mean ± SEM; WT: 1821 ± 115 pA, n = 6 animals; *crld-1(tm3993)*: 1826 ± 270 pA, n = 5 animals; p=0.6277, ns = not significant. Mann-Whitney test. (E) Response to pressure-ejection of nicotine in body wall muscles is unaffected in *crld-1(tm3993)*. Data indicate mean ± SEM; WT: 922 ± 76 pA, n = 4 animals; *crld-1(tm3993)*: 1289 ± 199 pA, n = 5 animals; p=0.1905, ns = not significant. Mann-Whitney test. (F) Confocal imaging of the L-AChR reporter UNC-29::tagRFP at the ventral nerve cords of wild-type and *crld-1(tm3993)* mutant adult worms. Scale bars = 10 μm. (G) Quantification of UNC-29::tagRFP fluorescence at the ventral nerve cords of wild-type and *crld-1(tm3993)* mutant adult worms. Data indicate mean ± SD; WT: n = 32 animals; *crld-1(tm3993)*: n = 32 animals; experiments were repeated three times.***p<0.001. Mann-Whitney test. (H) Confocal imaging of the GABA$_A$R reporter UNC-49::tagRFP at the ventral nerve cords of wild-type and *crld-1(tm3993)* mutant adult worms. Scale bars = 10 μm. (I) Quantification of UNC-49::tagRFP fluorescence at the ventral nerve cords of wild-type and *crld-1(tm3993)* mutant adult worms. Data indicate mean ± SD; WT: n = 31 animals; *crld-1(tm3993)*: n = 31 animals; experiments were repeated three times. p=0.4068, ns = not significant. Mann-Whitney test.

DOI: https://doi.org/10.7554/eLife.39649.006

To test if *crld-1* was required for the expression or function of other ligand-gated ion channels in muscle, we immuno-stained the GABA$_A$ receptor UNC-49 and found no difference in intensity and localization between *crld-1(tm3993)* mutants and the wild type (**Figure 3C**). To quantify GABA$_A$Rs, we used a knock-in strain in which TagRFP is fused to the N-terminus of the GABA$_A$R subunit UNC-49 and we found no difference between *crld-1* mutant and wild type (**Figure 3H,I**). Accordingly, responses to pressure-ejected GABA were indistinguishable between *crld-1(tm3993)* mutants and the wild type (**Figure 3D**). Similarly, the response to pressure-ejection of nicotine, which activates ACR-16-containing N-AChRs, was not significantly modified in *crld-1(tm3993)* mutants (**Figure 3E**).

Altogether these data showed that *crld-1* disruption impacts the expression of functional L-AChRs independently from NMJ formation but does not affect the expression of other ligand-gated ion channels at the NMJ.

## CRLD-1 stabilizes unassembled L-AChR subunits in the ER

The reduction of L-AChR at the NMJ of *crld-1(tm3993)* mutants might result from decreased synthesis or from intracellular retention of receptors. To distinguish between these hypotheses we quantified the overall amount of UNC-29 L-AChR subunit by western blot analysis. UNC-29 expression was decreased by approximately 50% in *crld-1(tm3993)* mutants as compared to the wild type (**Figure 4A**). To determine whether decreased L-AChR levels were due to defects at transcriptional or post-transcriptional steps, we quantified the levels of three mRNAs coding for L-AChR subunits and did not detect significant differences between the wild type and *crld-1(tm3993)* mutants (**Figure 4—figure supplement 1**).

Since CRLD-1 is an ER-resident protein, it might be involved in the assembly of the L-AChR subunits into a mature pentameric receptor or it might promote exit from the ER after assembly. To address this question, we analyzed UNC-29 levels in an *unc-63(kr13)* null mutant background: in the absence of the obligatory AChR subunit UNC-63, the remaining unassembled subunits, such as UNC-29, are retained in the ER and can be readily detected by western blot analysis (**Figure 4A**) (**Eimer et al., 2007**; **Richard et al., 2013**). Therefore, if CRLD-1 is required in the ER for stability and/or assembly of L-AChR subunits, UNC-29 levels will be decreased in a *crld-1;unc-63* double mutant as compared to the *unc-63* single mutant. By contrast, if CRLD-1 is required for the stability of L-AChRs after they exit the ER, UNC-29 levels will not be decreased in *crld-1;unc-63* animals. We found that UNC-29 levels are strongly decreased in *crld-1;unc-63* double mutant compared to single mutants, suggesting that CRLD-1 is important in the ER for the stability of unassembled subunits (**Figure 4A**).

To confirm that remaining receptors in *crld-1* mutants were properly trafficked to the Golgi after ER assembly, we analyzed the glycosylation patterns of UNC-29 using endoglycosidase H (EndoH) and N-glycosidase F (PNGaseF) (**Richard et al., 2013**). Nascent sugar side chains synthesized in the ER are cleaved by EndoH and PNGaseF, whereas mature N-glycosylations present on proteins that have already passed the *cis*-Golgi become resistant to EndoH. Consistently, UNC-29 subunits were EndoH sensitive in an *unc-63(kr13)* mutant background. By contrast, the digestion profile of UNC-29

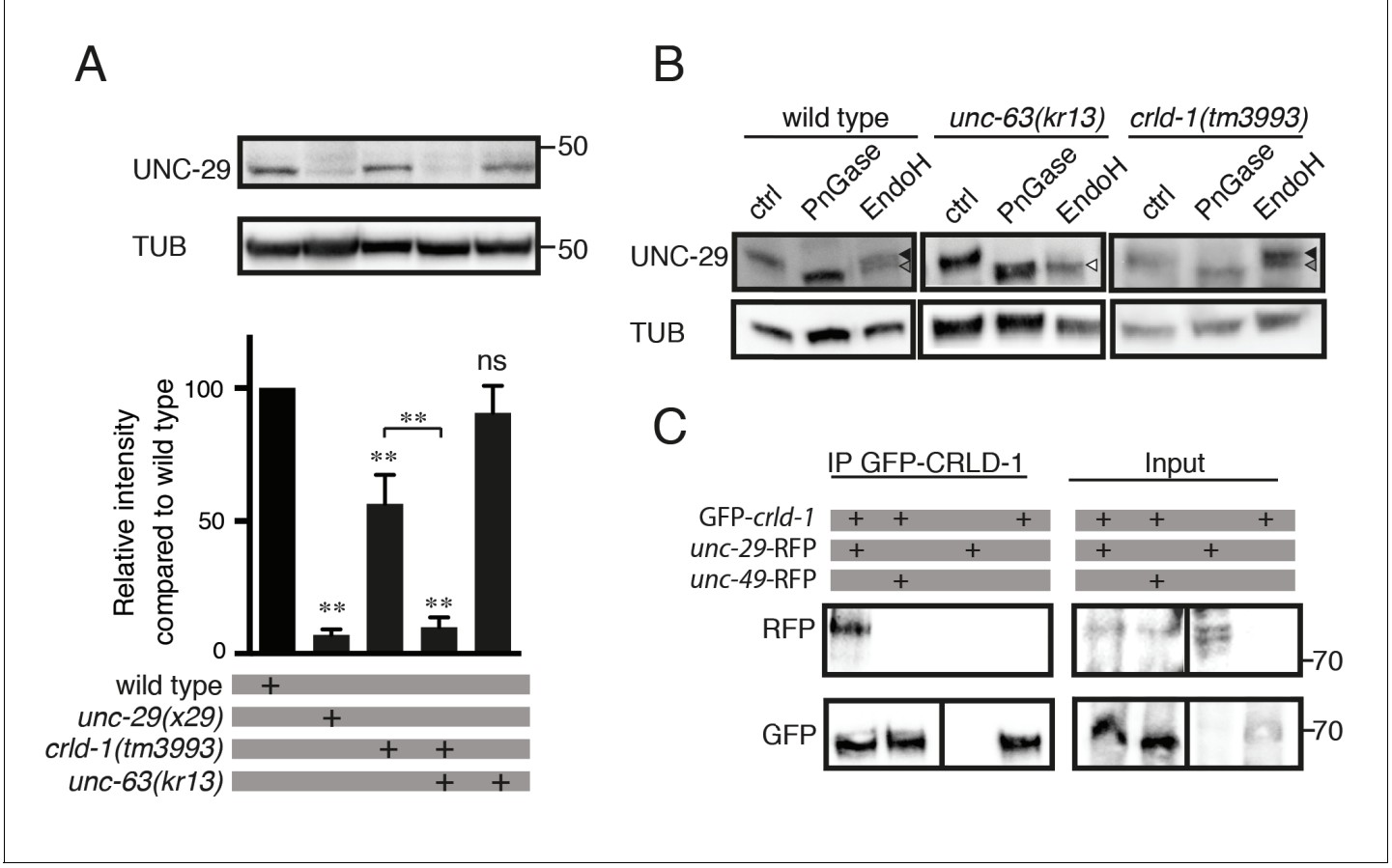

**Figure 4.** CRLD-1 is required for the stability of unassembled L-AChR subunits. (**A**) L-AChR expression is reduced in *crld-1(tm3993)* mutants. Levels of unassembled UNC-29 L-AChR subunits detected in *unc-63(kr13)* are further decreased in *unc-63(kr13);crld-1(tm3993)* double mutants. UNC-29 levels were quantified by western blot using anti-UNC-29 antibodies and normalized to tubulin levels. Significance is indicated compared to the wild type. The significance between *unc-63(kr13);crld-1(tm3993)* and *crld-1(tm3993)* is indicated by an horizontal line. Five independent experiments were quantified (mean ± SEM). p=0.6825, ns (not significant); **p=0.0079; after Bonferroni correction, Mann-Whitney test. TUB = tubulin. (**B**) Remaining L-AChRs exit the ER in *crld-1(tm3993)*. Treatments with EndoH or N-Glycosidase F (PNGase) were performed on protein extracts of mixed-stage animals before SDS-PAGE analysis. Black arrowheads indicate glycosylated forms resistant to EndoH, gray arrowheads indicate glycosylated forms partially resistant to EndoH, and white arrowheads indicate deglycosylated forms sensitive to EndoH. (**C**) CRLD-1 interacts with UNC-29 subunit of L-AChR in vivo. gfp-crld-1(kr298) animals were crossed with *rfp-unc-29(kr208)* or *rfp-unc-49(kr306)* to co-express CRLD-1 with RFP-tagged AChR or GABA_AR subunits, respectively. Immunoprecipitation of GFP-CRLD-1 using GFP-Trap beads co-immunoprecipitated RFP-UNC-29, but not UNC-49-RFP. As a control, GFP-CRLD alone was not immunoprecipitated by anti-RFP antibody. A vertical line indicates that the lanes are not adjacent in the gel.
DOI: https://doi.org/10.7554/eLife.39649.007

The following figure supplement is available for figure 4:

**Figure supplement 1.** – Measurement of L-AChR subunit mRNA levels.
DOI: https://doi.org/10.7554/eLife.39649.008

in *crld-1(tm3993)* was similar to the wild type, suggesting that the subunits that were not degraded in *crld-1(tm3993)* were properly assembled into functional receptors that could traffic to the plasma membrane (***Figure 4B***).

We then tested if CRLD-1 and L-AChRs could physically interact. Using immunoprecipitation experiments on total worm extracts, we found that endogenous GFP-CRLD-1 could indeed co-immunoprecipitate the RFP-tagged UNC-29 L-AChR subunit. By contrast, GFP-CRLD-1 could not immunoprecipitate the RFP-tagged UNC-49 GABA_A receptor (***Figure 4C***).

Altogether, these data suggested that CRELD-1 promotes L-AChR assembly in the ER through physical interaction with individual subunit(s) or L-AChR assembly intermediates.

## *C. elegans* CRLD-1 exhibits a PDI activity required for L-AChR assembly

It was recently shown that human CRELD2 is a putative protein disulphide isomerase (PDI) (*Hartley et al., 2013*). PDIs catalyze thiol-disulphide oxidation, reduction and isomerisation. They are critical for the correct formation of disulphide bonds or for the re-arrangement of incorrect bonds. These reactions involve CXXC amino-acid motifs in which cysteines are engaged in mixed disulphide complexes between the enzyme and the substrate. Mutation of the C-terminal cysteine in the active CXXC site generates a substrate trapping mutant by stabilizing covalent enzyme-substrate intermediate complexes (*Jessop et al., 2007*). Sequence comparison between human CRELD proteins and *C. elegans* CRLD-1 identified a conserved $C_{27}XXC_{30}$ motif in the N-terminal region of the CRLD-1 WE domain (*Figure 5A*).

Using a combination of genome engineering techniques, we introduced the C30A mutation in the previously generated *gfp-crld-1* knock-in (*Figure 5A*) to generate a potential substrate trapping mutant. GFP-CRLD-1(C30A) was ubiquitously expressed and displayed a localization pattern similar to the wild-type GFP-CRLD-1, yet its function was impaired based on the partial levamisole resistance of mutant animals (*Figure 5B* and *Figure 2—figure supplement 1C*). We then used a biochemical approach to test if CRLD-1 had a PDI-like activity in *C. elegans*. Total worm lysate proteins from both *gfp-crld-1(kr298)* (wild type) and *gfp-crld-1(kr302)* (C30A mutant) were separated by SDS-PAGE under reducing and non-reducing conditions. GFP-CRLD-1 was revealed by western blot analysis using an anti-GFP antibody. Under non-reducing conditions we detected high molecular weight species containing the mutated GFP-CRLD-1(C30A) that were absent in the non-mutated GFP-CRLD-1 (*Figure 5C*). These high-molecular weight complexes were resolved under reducing conditions. Altogether these data demonstrated that GFP-CRLD-1(C30A) behaved as a substrate trapping protein, strongly suggesting that CRLD-1 contains PDI activity required for proper synthesis of L-AChR.

## *Creld1* function is conserved across evolution

In mouse, the orthologous gene of *Crld-1a* is *Creld1*. To test for functional conservation between these two genes, we expressed a mouse *Creld1* cDNA in *C. elegans* body-wall muscle and found that the murine construct could rescue the levamisole sensitivity of the *crld-1(kr297)* null mutant (*Figure 6—figure supplement 1*). These data suggested that the function of *Creld1* was conserved across evolution and we decided to extend our analysis to mammalian systems.

*CRELD1* was reported to be expressed in human muscle tissue (*Rupp et al., 2002*). We confirmed the expression of CRELD1 proteins and transcripts in murine C2C12 myoblasts by Western blot and qPCR. These cells can be differentiated into myotubes and provide an established model to analyze AChR expression. We then stably transformed these cells with vectors expressing small hairpin RNAs (shRNAs) against mouse *Creld1* to achieve long-term knockdown of *Creld1* (*Figure 6A,B*). To test if CRELD1 was required for proper assembly of AChR in mouse muscle cells, we quantified the total amount and the surface fraction of AChRs after *Creld1* knock-down. Specifically, differentiated cells were incubated with biotin-α Bungarotoxin (α BT), receptors were then solubilized, pulled-down with streptavidin and quantified by western blot. As compared to control cells (*shScramble*), *shCreld1* caused a 50% decrease of AChR expression and further decreased the ratio of surface vs total AChR by 50% (*Figure 6A–E*). These changes were due neither to a transcriptional down-regulation of AChRα nor to a delay in differentiation since we found by qPCR that the mRNA levels of both *AChRα* subunit and *Myogenin* (marker of differentiation) were not affected by *Creld1* downregulation (*Figure 6F–G* and *Figure 6—figure supplement 2*). These results were highly consistent with what was found in the nematode and showed that CRELD1 is a limiting factor for AChR expression in mammals.

## Discussion

CRLD-1 was identified in a genetic screen for mutants partially resistant to the cholinergic agonist levamisole. This gene is evolutionarily conserved and our results show that the membrane-associated isoform of CRLD-1 (CRLD1-A in *C. elegans*, CRELD1 in mouse) is cell-autonomously required in both nematode and mammalian muscle cells to promote the surface expression of AChRs. We propose

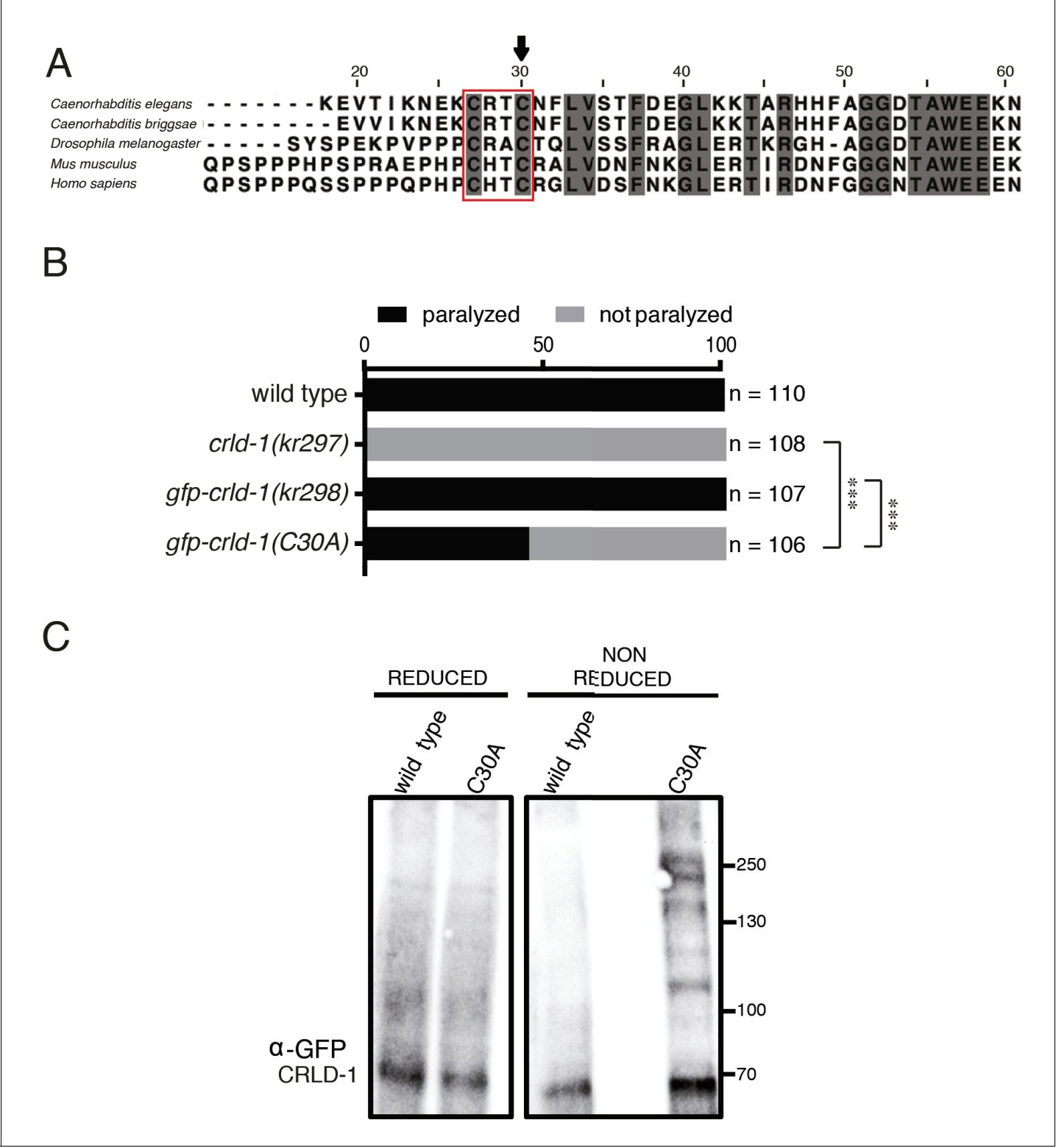

**Figure 5.** CRLD-1 displays putative PDI-like activity. (**A**) ClustalO alignment of *C. elegans* CRLD-1 with orthologous CRLD proteins from nematodes, fly, and vertebrates. The conserved CXXC motif present in the WE domain is boxed. Identical residues conserved in all species are highlighted in dark gray. The position of the cysteine residue that was mutated to generate the C30A mutation in *gfp-crld-1(C30A)* knock-in worms is indicated by an arrow. (**B**) Animals expressing the C30A mutation in the *crld-1* gene display partial levamisole resistance. Gray bars indicate the percentage of moving animals after overnight exposure to 1 mM levamisole, and black bars indicate the percentage of paralyzed animals. Experiments were repeated three

*Figure 5 continued on next page*

*Figure 5 continued*

times, n = number of animals tested. \*\*\*p<0.001, after Bonferroni correction, Fisher exact probability test. (C) The substrate-trapping CRLD-1 C30A mutant formed high molecular weight mixed disulphide complexes that were resolved under reducing conditions. In contrast, wild-type CRLD-1 did not form higher molecular weight complexes with putative substrate proteins. Total protein extract from *gfp-crld-1(kr298)* wild-type and *gfp-crld-1(kr302)* C30A mutant worms were separated by SDS-PAGE followed by Western blot analysis for GFP to detect CRLD-1.
DOI: https://doi.org/10.7554/eLife.39649.009

that this protein could act in the ER as a maturational enhancer to stabilize unassembled AChR subunits and to promote AChR assembly.

## CRLD-1A membrane topology

The transmembrane isoform CRLD-1A (ortholog of vertebrate CRELD1) contains two conserved transmembrane regions located at the C-terminus of the protein. Bioinformatic analysis of human CRELD1 suggests that both N- and C-termini reside in the extracellular spaces with a short intervening cytoplasmic loop (*Rupp et al., 2002*). In contrast, it was recently proposed that murine CRELD1 is localized at the ER membranes with the C- and N-termini facing the cytoplasm, based on differential sensitivity to proteases after partial cell permeabilization (*Mass et al., 2014*). Analysis of CRLD-1 in *C. elegans* suggests that CRLD-1A is an intrinsic ER-membrane protein with the C- and N-termini facing the ER lumen.

First, CRLD-1A and CRLD-1B isoforms are identical except for their short C-terminal regions (70 and 25 amino acids, respectively). Bioinformatic analyses identify a signal peptide at the N-terminus of CRLD-1, which predicts that the N-terminal region of CRLD-1 is translocated into the ER lumen. This is fully consistent with the localization of CRLD-1B, which behaves as a luminal ER protein. Second, over the course of our experiments, we overexpressed in muscle cells CRLD-1A fused to GFP at its N-terminus and we detected fluorescence at the plasma membrane of muscle cells, probably because some protein could escape the ER-retention machinery. In these transgenic worms, we injected fluorescently-labeled anti-GFP antibodies into the pseudo-coelomic cavity, a means to label cell-surface exposed epitopes (*Gottschalk and Schafer, 2006*) and we could stain the GFP at the muscle cell surface (MD, unpublished observation). Both results strongly suggest that the CRLD-1 N-terminal region, which represents most or the whole protein in CRLD-1A or −1B, respectively, localizes within the exoplasmic compartment. This is in full agreement with our results indicating the N-terminal region of CRLD-1 contains PDI activity, which most likely functions in the ER lumen. Since mouse *Creld1* cDNA rescues *C. elegans crld-1* mutants, it seems reasonable to propose that both mouse and nematode proteins have the same topology.

## CRLD-1 might act as a L-AChR maturational enhancer through chaperone and PDI functions

We found by substrate trapping experiments that CRLD-1 displays PDI activity. This PDI function is conserved through evolution, since human CRELD2 also has PDI activity. The PDI function of CRELD proteins relies on conserved CXXC motifs in the WE domain. These CXXC motifs are a common feature of thiol/disulphide oxidoreductases (*Hartley et al., 2013*). CRLD-1 has several CXXC motifs but we selected a conserved amino-terminal CXXC motif to generate a substrate-trapping mutant, similar to those characterized for mammalian *Creld2* (*Hartley et al., 2013*). The CXXA mutation does not totally impair the function of CRLD-1, although the mutant protein behaves as a *bona fide* substrate-trapping protein based on biochemical experiments. The residual activity of the protein could be explained by the activity of other CXXC sites present in the CRLD-1 protein. However, we do not favor this hypothesis because the CXXC motifs that are present more C-terminally localize in predicted EGF domains and the cysteines are likely engaged in structural disulphide bonds that stabilize EGF domains. Consistently, Hartley et al. demonstrated that the carboxy-terminal CXXC motifs of CRELD2 do not possess isomerase activity. Therefore, the residual activity of CXXA CRLD-1 mutants might indicate that CRLD-1 also behaves as a chaperone that stabilizes partially assembled AChRs, as shown for other PDIs (*Hatahet and Ruddock, 2007*).

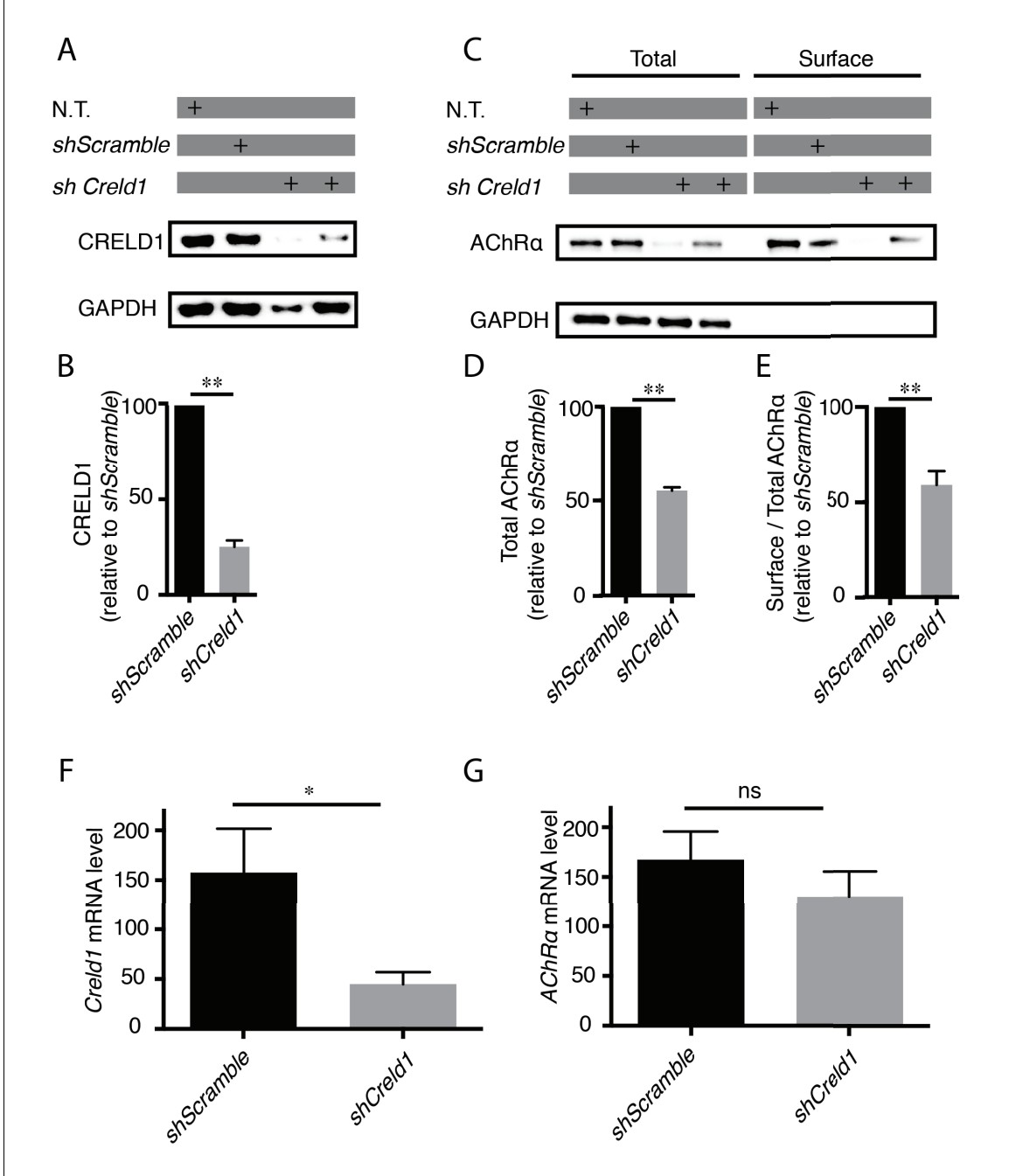

**Figure 6.** *Creld1* knockdown leads to reduced AChR expression at the plasma membrane *in vitro*. (A–E) Measurement of mouse CRELD1 and AChR α subunit protein levels. C2C12 cells expressing shRNA against *Creld1* (*shCreld1*) or scrambled (*shScramble*) sequences were differentiated for 5 days and then subjected to surface labeling with αBT-biotin for AChRα. N.T. = non transfected cells. Streptavidin precipitates (surface) and total lysates were separated by SDS/PAGE and probed for indicated proteins (A and C). CRELD1 protein levels were reduced by 75% in cells expressing *shCreld1* as compared to *shScramble* (B). Quantitation of total AChRα levels (D) and of the surface to total AChRα ratio (E) in *shScramble* (100%) and *shCreld1* cells from n = 5 independent experiments. Error bars, SEM; \*\*p=0,0079, Mann-Whitney test. (F–G) Measurement of mouse *Creld1* and *AChRα* subunit mRNA levels. C2C12 cells expressing *shScramble* or *shCreld1* were differentiated for 5 days and then subjected to RNA extraction. Quantitative real-time PCR measurements of mRNA levels for *Creld1* (F) and *AChRα* subunit (G). *Creld1* mRNA is decreased in *shCreld1* cells compared to *shScramble* cells, whereas *AChRα* subunit mRNA is not significantly decreased in *shCreld-1* cells. Mean ± SEM is shown in six independent experiments. \*p=0,0411; p=0,4740, ns (not significant), Mann–Whitney test.

DOI: https://doi.org/10.7554/eLife.39649.010

The following figure supplements are available for figure 6:

*Figure 6 continued on next page*

*Figure 6 continued*

**Figure supplement 1.** Mouse *Creld1* rescues levamisole-sensitivity of *crld-1* mutant worms.
DOI: https://doi.org/10.7554/eLife.39649.011

**Figure supplement 2.** Knocking down *Creld1* does not impact *Myogenin* transcriptional levels.
DOI: https://doi.org/10.7554/eLife.39649.012

## CRELD1 and CRELD2 are evolutionarily-conserved ER resident proteins

CRELD proteins are highly conserved across species. Similarity is not restricted to the genetically mobile EGF domains but also to the conserved WE domain, which contains a high content of tryptophan and glutamic acid residues (*Rupp et al., 2002*). Most vertebrate genomes contain two separate paralogous genes, *Creld1* and *Creld2*. CRELD1 is an integral membrane protein containing two transmembrane segments in its carboxy-terminal region, while CRELD2 is secreted in the exoplasmic compartment and contains the ER-retention motif RDEL at its C-terminus. In *C. elegans,* and likely in *Drosophila*, there is only one gene, *crld-1*, coding for both CRELD versions. Two transcripts are generated by alternative splicing of the last exons and code for two proteins, namely CRLD-1A, which ends with two transmembrane domains and is similar to vertebrate CRELD1, and CRLD-1B, which ends with a KDEL sequence and is similar to vertebrate CRELD2.

Although the two CRLD-1 isoforms are nearly identical and expressed in the same cells in *C. elegans* we demonstrated that they are not redundant. First, the two isoforms localize in the ER in muscle cells, yet CRLD-1A has a more pronounced perinuclear localization and CRLD-1B has a more punctate pattern, in agreement with the predicted localization of CRLD-1A in the ER membrane and CRLD-1B in the ER lumen. Second, the transmembrane isoform CRLD-1A is the only isoform necessary and sufficient to regulate L-AChR biogenesis in muscle cells. CRLD-1A can interact, directly or indirectly, with AChR subunits in the ER based on co-immunoprecipitation experiments. This might involve an interaction of the transmembrane regions of CRLD-1A with AChR subunits. Alternatively, targeting the luminal domain of CRLD-1 to the ER membrane might favor the interaction with AChR subunits by increasing the apparent concentration of CRLD-1 at the membrane. It is also possible that CRLD-1A might be recruited to specific chaperoning domains of the ER membrane where AChRs are assembled. Interestingly, CRELD2 was suggested to behave as a negative regulator of α4β2 AChR expression during nicotine-induced up-regulation (*Hosur et al., 2009*). Whether CRELD1 and CRELD2 have antagonistic functions in some cellular contexts remains to be investigated.

## Specific requirement of general protein synthesis factors for L-AChR biogenesis

From this work, CRLD-1 seems to support a very specific function since its disruption severely impairs the expression of the heteromeric L-AChR but does not affect the synthesis of homomeric N-AChRs nor of GABA$_A$Rs. However, CRLD-1 is also expressed in *C. elegans* cells that do not synthesize L-AChRs. It is therefore extremely likely that CRLD-1 is involved in the biogenesis of additional proteins that remain to be identified. Since *C. elegans crld-1* null mutants are viable and display no obvious abnormal phenotypes, it suggests that pathways redundant to CRLD-1 can compensate for *crld-1* inactivation, maybe by using other members of the PDI family. If this is the case, the apparent specificity of CRLD-1 for L-AChR synthesis would rather arise from the intrinsic characteristics of AChR folding and assembly. Because these steps were shown in other species to be slow and inefficient, defects in factors required in the ER for protein biogenesis might be more difficult to compensate for in the case of L-AChRs, hence providing justification to use L-AChR expression as a sensitive proxy to identify new components along its biogenesis pathway.

Accordingly, previous genetic screens in *C. elegans* identified auxiliary proteins absolutely required for receptor biosynthesis, namely RIC-3, UNC-50, and UNC-74. RIC-3 is required for assembly of all AChRs in the ER, including L-AChRs and N-AChRs (*Boulin et al., 2008*; *Halevi et al., 2002*; *Jospin et al., 2009*). It is conserved in flies and mammals and can either promote or inhibit the expression of AChRs and 5-HT$_3$ receptors in heterologous systems (*Millar and Harkness, 2008*). UNC-50 is orthologous to GMH1, a protein conserved from yeast to humans, which interacts with a guanine nucleotide exchange factor of the small G protein Arf. In *C. elegans* UNC-50 localizes to the Golgi and promotes the targeting of L-AChRs to the plasma membrane, thereby preventing their

degradation in lysosomes (*Abiusi et al., 2017*; *Eimer et al., 2007*). UNC-74 is a predicted thiore-doxin homologous to TMX-3 (*Boulin et al., 2008*). It is necessary for L-AChR expression, but its detailed function remains uncharacterized. Screens for mutants with only partially decreased sensitivity to levamisole also identified the gene *emc-6* that is required for the assembly of AChR but also GABA$_A$ receptor subunits (*Richard et al., 2013*). The EMC-6 protein is part of the EMC complex. Initially identified in yeast (*Jonikas et al., 2009*), the EMC has been extremely well conserved throughout evolution (*Wideman, 2015*). Interestingly, EMC subunits were shown to be required in the ER for the synthesis of multi-pass transmembrane proteins in *Drosophila* (*Satoh et al., 2015*), suggesting that the EMC might be chaperoning transmembrane proteins at early steps. A recent study performed in both yeast and human cells demonstrates that the EMC complex initiates client interaction cotranslationally and remains associated after completion of translation. This prevents premature degradation and promotes recruitment of substrate-specific and general chaperones (*Shurtleff et al., 2018*). We can hypothesize that CRLD-1A acts downstream the EMC complex to assist the maturation of the L-AChR in the ER.

## CRELD1 displays a conserved function in the regulation of AChR

All the genes identified in *C. elegans* as being involved in L-AChR synthesis are conserved in mammals, yet their requirement for AChR biogenesis has not been systematically tested. RIC-3 mostly regulates neuronal α7 AChRs and might interfere with α4β2 AChR expression (*Alexander et al., 2010*; *Dau et al., 2013*). A mutation in the human gene *UNC50* was recently associated with arthrogryposis, a severe fetal disease that can be caused by impairment of neurotransmission at the NMJ. This suggested that the UNC-50 ortholog might also be required in humans for muscle AChR expression (*Abiusi et al., 2017*). CRLD-1A has been widely conserved during evolution and here we demonstrate that its function was also conserved. First, murine *Creld1* controls the expression of AChR in mouse muscle cells in a very similar way as in *C. elegans*. Second, the mouse *Creld1* gene can rescue the defects of *crld-1* mutants, suggesting that the molecular mechanisms that we analyzed in detail in nematodes are relevant for CRELD1 function in mammals. Mutations in *CRELD1*, the human ortholog of CRLD-1A, are linked to atrioventricular septal defects, which represent more than 7% of all congenital heart defects in human. The molecular mechanisms that we identified may trigger novel research directions to elucidate the physiopathology of these diseases.

Altogether, our results indicate that the early steps of AChR biogenesis rely on factors that could ultimately be targeted to modify AChR expression without altering the entire protein biosynthesis machinery. Hence, CRELD1 potentially represents a novel target to modulate AChR levels in pathological contexts such as congenital myasthenic syndromes and possibly chronic exposure to nicotine, which causes increased AChR expression in the brain of cigarette smokers.

## Materials and methods

**Key resources table**

| Reagent type (species) or resource | Designation | Source or reference | Identifiers | Additional information |
|---|---|---|---|---|
| Gene (*C. elegans*) | *crld-1*, F09E8.2 | this paper | WormBase ID: WBGene00 008624 | See Results, Disruption of the evolutionarily conserved gene crld-1 confers partial resistance to the cholinergic agonist levamisole. |
| Gene (*M. musculus*) | *Creld1*, cysteine rich with EGF like domains 1 | PMID:12137942, PMID: 25328912, PMID: 24697899 | GeneID: 171508 | |

*Continued on next page*

*Continued*

| Reagent type (species) or resource | Designation | Source or reference | Identifiers | Additional information |
|---|---|---|---|---|
| Strain, strain background (*C. elegans*) | EN13 | doi:10.1534 /genetics. 104.038265 | WormBase ID:WBVar 00088264 | Strain background: N2 |
| Strain, strain background (*C. elegans*) | ZZ29 | PMID:3668616 | WormBase ID: WBV ar00275223 | Strain background: N2 |
| Strain, strain background (*C. elegans*) | EN208 | PMID:24896188 | WormBase ID: WBV ar02125731 | Strain background: N2 |
| Strain, strain background (*C. elegans*) | CB407 | PMID:10377345 | WormBase ID:WBVar00143186 | Strain background: N2 |
| Strain, strain background (*C. elegans*) | EN296: *unc-49(kr296:: tagRFP)* | this paper | | Strain background: N2. See M and M, Strains and Genetics |
| Strain, strain background (*C. elegans*) | EN132: *creld-1 (kr132::Mos1)* | this paper | | Strain background: N2. See M and M, Strains and Genetics |
| Strain, strain background (*C. elegans*) | EN133: *crld-1 (kr133::Mos1)* | this paper | | Strain background: N2. See M and M, Strains and Genetics |
| Strain, strain background (*C. elegans*) | EN2169: *crld-1(tm3993)* | PMID:23173093 | WormBase ID: WBVar00 252554 | Strain background: N2. See M and M, Strains and Genetics |
| Strain, strain background (*C. elegans*) | EN297: *crld-1 (kr297::HySOG)* | this paper | | Strain background: N2. See M and M, Strains and Genetics |
| Strain, strain background (*C. elegans*) | EN298: *crld-1 (kr298::GFP)* | this paper | | Strain background: N2. See M and M, Strains and Genetics |
| Strain, strain background (*C. elegans*) | EN302: *crld-1 (kr302::GFPC30A)* | this paper | | Strain background: N2. See M and M, Strains and Genetics |
| Strain, strain background (*C. elegans*) | EN303: *crld-1b (kr303::GFP)* | this paper | | Strain background: N2. See M and M, Strains and Genetics |
| Strain, strain background (*C. elegans*) | EN308: *crld-1a (kr308::GFP)* | this paper | | Strain background: N2. See M and M, Strains and Genetics |

*Continued on next page*

*Continued*

| Reagent type (species) or resource | Designation | Source or reference | Identifiers | Additional information |
|---|---|---|---|---|
| Strain, strain background (*C. elegans*) | EN2097: *crld-1 (tm3993);unc-29 (kr208::tagRFP)* | this paper | | Strain background: EN2169, EN308. See M and M, Strains and Genetics |
| Strain, strain background (*C. elegans*) | EN4059: *crld-1 (tm3993);unc-49 (kr296::tagRFP)* | this paper | | Strain background: EN2169, EN296. See M and M, Strains and Genetics |
| Strain, strain background (*C. elegans*) | EN2544: *crld-1 (tm3993); krEx870 [Pmyo-3::crld-1b cDNA; myo-2::gfp]* | this paper | | Strain background: EN2169. See M and M, Strains and Genetics |
| Strain, strain background (*C. elegans*) | EN2545: *crld-1(tm3993); krEx870[Pmyo-3 ::crld-1b cDNA; myo-2::gfp]* | this paper | | Strain background: EN2169. See M and M, Strains and Genetics |
| Strain, strain background (*C. elegans*) | EN2546: *crld-1(tm3993); krEx870[Pmyo-3:: crld-1b cDNA; myo-2::gfp]* | this paper | | Strain background: EN2169. See M and M, Strains and Genetics |
| Strain, strain background (*C. elegans*) | EN2548: *crld-1(tm3993); krEx871[Pmyo-3:: crld-1a cDNA; myo-2::gfp]* | this paper | | Strain background: EN2169. See M and M, Strains and Genetics |
| Strain, strain background (*C. elegans*) | EN2549: *crld-1(tm3993); krEx871[Pmyo-3:: crld-1a cDNA; myo-2::gfp]* | this paper | | Strain background: EN2169. See M and M, Strains and Genetics |
| Strain, strain background (*C. elegans*) | EN2550: *crld-1(tm3993); krEx871[Pmyo-3:: crld-1a cDNA; myo-2::gfp]* | this paper | | Strain background: EN2169. See M and M, Strains and Genetics |
| Strain, strain background (*C. elegans*) | EN3790:*crld-1(kr297); krEx1277[Pmyo-3:: mouse-creld-1 cDNA; myo-2::gfp]* | this paper | | Strain background: EN297. See M and M, Strains and Genetics |
| Strain, strain background (*C. elegans*) | EN3791:*crld-1 (kr297); krEx1277 [Pmyo-3::mouse-creld-1 cDNA; myo-2::gfp]* | this paper | | Strain background: EN297. See M and M, Strains and Genetics |

*Continued on next page*

*Continued*

| Reagent type (species) or resource | Designation | Source or reference | Identifiers | Additional information |
|---|---|---|---|---|
| Strain, strain background (C. elegans) | EN3793: *crld-1(kr297); krEx1277[Pmyo-3:: mouse-creld-1 cDNA; myo-2::gfp]* | this paper | | Strain background: EN297. See M and M, Strains and Genetics |
| Strain, strain background (C. elegans) | EN1700: *crld-1(kr132: :Mos1);krEx456 [pTB208; punc-122::gfp]* | this paper | | Strain background: EN132. See M and M, Strains and Genetics |
| Strain, strain background (C. elegans) | EN3465: *crld-1b(kr303:: GFP);krEx1245 [Pmyo-3::MANS:: TagRFP-T (pMR61), rol-6(su1006, panneuronal DsRed2 (pCB101)]* | this paper | | Strain background: EN303. See M and M, Strains and Genetics |
| Strain, strain background (C. elegans) | EN3504: *crld-1b(kr303::GFP); krEx1250 [Pmyo-3:: tagRFP-T::KDEL (pMR68), rol-6 (su1006, panneuronal DsRed2 (pCB101)]* | this paper | | Strain background: EN308. See M and M, Strains and Genetics |
| Strain, strain background (C. elegans) | EN3478: *crld-1a(kr308::GFP); krEx1246 [Pmyo-3::MANS:: TagRFP-T (pMR61), rol-6(su1006, panneuronal DsRed2 (pCB101)]* | this paper | | Strain background: EN308. See M and M, Strains and Genetics |
| Strain, strain background (C. elegans) | EN3501: *crld-1a(kr308:: GFP);krEx1249 [Pmyo-3:: TagRFP-T::KDE;, rol-6(su1006) panneuronal DsRed2]* | this paper | | Strain background: EN308. See M and M, Strains and Genetics |
| Cell line (M. musculus) | C2C12 mouse myoblasts | | RRID:CVCL_0188 | |
| Recombinant DNA reagent | MSF037586 -3-CU6(OS262215) shRNA against mouse Creld1 (plasmid) | this paper | GeneCopoeia | (gccttggctac tttgaggc) See M and M, Cell Culture and Western Blot |
| Antibody | anti-UNC-38 | PMID: 19794415 | | (1:500) |
| Antibody | anti-UNC-49 | PMID: 12684444 | | (1:500) |
| Antibody | anti-VAChT/ UNC-17 | PMID: 15457263 | | (1:1000) |
| Antibody | Cy3-labeled goat anti-rabbit | Jackson Immuno Research Laboratories | | (1:1000) |

*Continued on next page*

*Continued*

| Reagent type (species) or resource | Designation | Source or reference | Identifiers | Additional information |
|---|---|---|---|---|
| Antibody | A488-labeled goat anti-mouse | Molecular Probes | Cat. No.: A32723 | (1:500) |
| Antibody | A488-labeled goat anti-rat | Molecular Probes | Cat. No.: A-11006 | (1:1000) |
| Antibody | anti-Creld1 | Abcam | Cat. No.: ab140346 | (1:500) |
| Antibody | purified mouse Anti-Acetylcholine Receptor alpha | BD Transduction Laboratories | Cat. No.: 610989 | (1:500) |
| Antibody | anti GAPDH | Merck | Cat. No.: MAB374 | (1:10000) |
| Antibody | Goat anti-Mouse IgG (H + L) Secondary Antibody, HRP | Thermo Fisher Scientific | Cat. No.: 62–6520 | (1:3000) |
| Antibody | Goat anti-Rabbit IgG (H + L) Secondary Antibody, HRP | Thermo Fisher Scientific | Cat. No.: 65–6120 | (1:3000) |
| Antibody | anti-UNC-29 | PMID:23431131 | | (1:1000) |
| Antibody | anti-TUBULIN | Sigma | Cat. No.:T9026-2ML | (1:1000) |
| Antibody | mouse anti-GFP | Roche | Cat. No.: 1181 4460001 | (1:1000) |
| Antibody | RFP mouse monoclonal | ThermoFisher Scientific | Cat. No.:MA5-15257 | (1:1000) |
| Recombinant DNAreagent | pTB205: *Pmyo-3:: crld-1a cDNA* | this paper | | See M and M, Strains and Genetics |
| Recombinant DNA reagent | pTB206: *Pmyo-3:: crld-1b cDNA* | this paper | | See M and M, Strains and Genetics |
| Recombinant DNA reagent | pTB208: 4,6 kb genomic fragment containing crld-1 and upstream regulatory regions fused to SL2-GFP | this paper | | See M and M, Strains and Genetics |
| Recombinant DNA reagent | pMR61: *Pmyo-3::RFP::MANS* | PMID:23431131 | | |
| Recombinant DNA reagent | pMR68: *Pmyo-3::RFP::KDEL* | PMID:23431131 | | |
| Recombinant DNA reagent | pMD20: *Pmyo-3:: mouse-creld-1 cDNA.* | this paper | | See M and M, Strains and Genetics |
| Recombinant DNA reagent | pMD1: *Pcrld-1:: HySOG-crld-1:: unc-54 3'UTR.* | this paper | | See M and M, Strains and Genetics |

*Continued on next page*

*Continued*

| Reagent type (species) or resource | Designation | Source or reference | Identifiers | Additional information |
|---|---|---|---|---|
| Recombinant DNA reagent | pHZ34: *Pcrld-1 ::GFP-creld-1:: unc-54 3'UTR.* | this paper | | See M and M, Strains and Genetics |
| Recombinant DNA reagent | pMD3: this plasmid was created on the basis of pHZ34; the C30A point mutation (TGC > GCT) was introduced in the sequence of crld-1 gene. | this paper | | See M and M, Strains and Genetics |
| Recombinant DNA reagent | pMD5: (*1 st I-SceI sgRNA*) | this paper | | See M and M, Strains and Genetics |
| Recombinant DNA reagent | pMD7: (*2nd I-SceI sgRNA*) | this paper | | See M and M, Strains and Genetics |
| Recombinant DNA reagent | pPT02 | PMID: 28280212 | | |
| Recombinant DNA reagent | pMD8: (*dpy-10 sgRNA*) | PMID:25161212, this paper | | See M and M, Strains and Genetics |
| Recombinant DNA reagent | pMD10: (*exon9a sgRNA*) | this paper | | See M and M, Strains and Genetics |
| Recombinant DNA reagent | pMD11: (*exon9b sgRNA*) | this paper | | See M and M, Strains and Genetics |
| Commercial assay or kit | TURBO DNA free kit, Ambion | Fisher, | Cat. No.:AM1907 | |
| Commercial assay or kit | iScript cDNA synthesis kit | BioRad | Cat. No.:1708891 | |
| Chemical compound, drug | αBT-biotin | Molecular probes/fisher | Cat. No.:B1196 | |
| Chemical compound, drug | Streptavidin (Sepharose Bead Conjugate) | Cell Signaling technology/ ozyme | Cat. No.:3419S | |
| Chemical compound, drug | levamisole | Sigma | Cat. No.:L9756 (10G) | |

## Strains and genetics

*C. elegans* strains were cultured as described previously (*Brenner, 1974*) and kept at 20°C, unless indicated otherwise. The following mutations were used in this study: LG I: *unc-29(x29)*, *unc-63 (kr13)*; LG IV: *crld-1(kr132, kr133, tm3993, kr297)*.

Strains, expression constructs, transgenic animals and generation of knock-in worms are listed and described below.

## List of strains

The following mutant alleles and transgenes were used in this study:

LGI: *unc-63(kr13)*, *unc-29(x29)*, *unc-29(kr208::tagRFP)* (*Richard et al., 2013*);

LGIII: *unc-49(e407); unc-49(kr296::tagRFP)*

LGIV: *creld-1(kr132::Mos1), crld-1(kr133::Mos1), crld-1(tm3993); crld-1(kr297::HySOG), crld-1 (kr298::GFP), crld-1(kr302::GFP-C30A), crld-1b(kr303::GFP), crld-1a(kr308::GFP)*;

LGV: *acr-16(ok789)*.

The following transgenic lines were created for this study:

Extrachromosomal array in *crld-1(tm3993)*: *krEx870[Pmyo-3::crld-1b cDNA; myo-2::gfp]*, *krEx871 [Pmyo-3::crld-1a cDNA; myo-2::gfp]*.

Extrachromosomal array in *crld-1(kr297)* : *krEx1277[Pmyo-3::mouse-creld-1 cDNA; myo-2::gfp]*.

Extrachromosomal array in *crld-1(kr132 ::Mos1)*: *krEx456[pTB208; punc-122::gfp]*.

Extrachromosomal array in *crld-1b(kr303::GFP)*: *krEx1245 [Pmyo-3::MANS::TagRFP-T (pMR61), rol-6(su1006, panneuronal DsRed2 (pCB101)]*, *krEx1250 [Pmyo-3::tagRFP-T::KDEL (pMR68), rol-6 (su1006, panneuronal DsRed2 (pCB101)]*.

Extrachromosomal array in *crld-1a(kr308::GFP)*: *krEx1246 [Pmyo-3::MANS::TagRFP-T (pMR61), rol-6(su1006, panneuronal DsRed2 (pCB101)]*, *krEx1249 [Pmyo-3::TagRFP-T::KDEL (pMR68), rol-6 (su1006, panneuronal DsRed2 (pCB101)]*.

## *C. elegans* germline transformation

Transformation was performed by microinjection of DNA mixture in the gonad of young adults. The total DNA concentration of the injection mix was normalized at 100 ng/μL using 1kb + ladder (Invitrogen). The following plasmids were used for *C. elegans* germline transformation:

- pTB205: *Pmyo-3::crld-1a* cDNA
- pTB206: *Pmyo-3::crld-1b* cDNA
- pTB208: 4,6 kb genomic fragment containing *crld-1* and upstream regulatory regions fused to SL2-GFP
- pMR61: *Pmyo-3::RFP::MANS* (**Richard et al., 2013**)
- pMR68: *Pmyo-3::RFP::KDEL* (**Richard et al., 2013**)
- pMD20: *Pmyo-3::mouse-creld-1 cDNA*.

## Generation of deletion and single-copy insertion alleles by MegaTIC

The final *gfp-crld-1(kr298)* knock-in was generated using the MegaTIC technique, this protocol consists of 2 steps (Ji, T., Ibanez-Cruceyra, P., D'Alessandro, M., Bessereau JL, in preparation).

In the first step, the *crld-1(kr297)* molecular null allele was generated by using the *Mos*TIC technique as previously described (**Robert et al., 2009**). 49 nucleotides coding for *crld-1* and starting from the ATG of *crld-1* were replaced by the HySOG cassette, that contains both positive (hygromycin B) and negative (miniSOG, a fluorescent protein engineered to produce singlet oxygen upon blue light illumination) selection markers flanked by two meganuclease I-SceI target sites. The pMD1 vector was injected as a rescue template into a strain containing the *kr133 Mos1* insertion in the fourth exon of *crld-1* gene. In pMD1, a 1,5-kilobase (kb) left *crld-1* homology sequence and a 4 kb right homology sequence flank the HySOG cassette. *kr297* knock-in allele was identified using positive selection of worms containing the HySOG cassette, therefore selecting worms resistant to hygromycin B.

In the second step, the HySOG cassette was excised by meganuclease-induced chromosomal breaks on each I-Sce-I site in the presence of pHZ34 as a repair template. pHZ34 contains gfp fused to the 5' of the *crld-1* gene. In pHZ34, a 1,5-kilobase (kb) left *crld-1* homology sequence and a 4 kb right homology sequence flank the *gfp. Gfp-crld-1(kr298)* knock-ins were identified based on their resistance to blue light illumination, followed by PCR analysis.

## Generation of single-copy insertion mutant allele by combining Co-CRISPR and MegaTIC

The *gfp-crld-1(kr302)* knock-in containing the C30A point mutation was generated using a combination of Co-CRISPR and MegaTIC techniques. The starting point was the excision of the HySOG cassette from the *crld-1(kr297)* by using sgRNAs against each I-Sce-I site and in presence of the pMD3 repair template. pMD3 contains *gfp* fused to the 5' of the *crld-1* gene and the mutation TGC >GCT (C30A point mutation). In pMD3, a 1,5-kilobase (kb) left *crld-1* homology sequence and a 4 kb right homology sequence flank the *gfp*. We chose *dpy-10* as a co-conversion marker to introduce a

dominant mutation causing a visible Rol/Dpy phenotype(**Arribere et al., 2014**). Rol/Dpy F1 worms were preselected for negative selection by blue light illumination and confirmed by PCR analysis.

## Generation of single-copy insertion mutant allele by Co-CRISPR

The *gfp-crld-1(kr298)* knock-in was injected with sgRNAs against the splicing acceptor site of either exon 9a or exon 9b of *crld-1* gene. Linear repair templates with short (≈ 30–40 bases) homology arms (**Paix et al., 2014**) were injected as a rescue template into *gfp-crld-1(kr298)*. The linear repair templates contained PmeI restriction site followed by a STOP codon in place of the AG splicing acceptor site of either exon 9a or exon 9b. *Dpy-10* was used as a co-conversion marker to introduce a dominant mutation causing a visible Rol/Dpy phenotype (**Arribere et al., 2014**). Engineered worms were identified by PCR.

The following plasmids were used for generation of deletion and single-copy insertion alleles:

- pCFJ601: *Peft-3::Mos1 transposase::tbb-2 3'UTR*, Addgene #34874.
- pMA122: *Phsp16-41::peel-1::tbb-2 3'UTR*. Addgene #34873.
- pDD162: *Peft-3::Cas-9::tbb-2 3'UTR*, Addgene #47549.
- pPD118.33: *Pmyo-2::GFP*.
- pHZ34: *Pcrld-1::GFP-creld-1::unc-54 3'UTR*. *crld-1* promoter and gene were amplified from genomic DNA and, by PCR fusion, *gfp* was inserted before the first exon of *crld-1*.
- pMD1: *Pcrld-1::HySOG-crld-1::unc-54 3'UTR*. A fragment of 3379 nucleotides containing HySOG (hygromycinB miniSOG dual selection cassette) flanked on each side by I-SceI sites, was inserted in pHZ34 vector between PstI and Bsp1407I sites. 763 nucleotides, encompassing the first exon of *crld-1* gene fused to the *gfp* gene, were removed. Different fragments were assembled using isothermal assembly (**Gibson et al., 2009**).
- pMD3: this plasmid was created on the basis of pHZ34. Using Gibson cloning the C30A point mutation (TGC >GCT) was introduced in the sequence of *crld-1* gene.
- pMD5 (1 st I-SceI sgRNA): This vector was created on the basis of pPT02(**El Mouridi et al., 2017**). The pPT02 vector contains a *C. elegans* U6 promoter and 3' UTR (based on **Friedland et al., 2013**) and two restriction sites (PmeI and SexAI) to linearize the vector, followed by the invariant sgRNA scaffold sequence (T. Boulin lab.). To generate the sgRNA expression vector pMD5, the protospacer sequence was inserted between the U6 promoter and the sgRNA scaffold, using PmeI and SexAI sites of pPT02. The protospacer contained the sequence of the 5' I-Sce-I site flanking the HySOG cassette in pMD1 and was synthetized by Sigma: AATTGCAAATCTAAATGTTTgACCCTGCAGGTAGGGATAACGTTTTAGAGCTAGAAATAGC.
- pMD7 (2nd I-SceI sgRNA): This vector was created on the basis of pPT02. The protospacer sequence was inserted between the U6 promoter and the sgRNA scaffold, using PmeI and SexAI sites of pPT02. The protospacer contained the 3' I-Sce-I site flanking the HySOG cassette in pMD1 and was synthetized by Sigma: AATTGCAAATCTAAATGTTTgAGGGATAACAGGGTAATCGCGTTTTAGAGCTAGAAATAGC
- pMD8 (*dpy-10* sgRNA): This vector was created on the basis of pPT02. The protospacer sequence was inserted between the U6 promoter and the sgRNA scaffold, using PmeI and SexAI sites of pPT02. The protospacer was synthetized by Sigma: AATTGCAAATCTAAATGTTTgCTACCATAGGCACCACGAGGTTTTAGAGCTAGAAATAGC
- pMD10 (exon9a sgRNA): this vector was created on the basis of pPT02. The protospacer sequence was inserted between the U6 promoter and the sgRNA scaffold, using PmeI and SexAI sites of pPT02. The protospacer was synthetized by Sigma:

AATTGCAAATCTAAATGTTTGATCAGGAGATGCTGAACCAGTTTTAGAGCTAGAAATAGC

The linear templates used as repair template contained PmeI restriction site (GTTTAAC) followed by a STOP codon (TAA) in place of the AG splicing acceptor site of exon 9a and was also synthetized by Sigma:
CACCCAGTTCCAATTTCCTCTATTCACCATGGTTCGTTTAAACTAACATCTCCTGATCGCCCG TTCATGCCAATCGACCAGC

- pMD11 (exon9b sgRNA): this vector was created on the basis of pPT02. The protospacer sequence was inserted between the U6 promoter and the sgRNA scaffold, using PmeI and SexAI sites of pPT02. The protospacer was synthetized by Sigma:

GCTATTTCTAGCTCTAAAACTTGCTTTCAAGGCTGCAAATcAAACATTTAGATTTGCAATT

The linear templates used as repair template contained PmeI restriction site (GTTTAAC) followed by a STOP codon (TAA) in place of the AG splicing acceptor site of exon 9b and was also synthetized by Sigma:
CAATAAAATGTAGAATATTTTCATTTTTTCAAAATTTGCGTTTAAACTAACCTTGAAAGCAACG-GAACAGCAAGCTCATGAAGATG

## Levamisole assay

Assays for levamisole sensitivity after overnight exposure were performed as previously described in (*Richard et al., 2013*). Tetramisole hydrochloride (Sigma-Aldrich) was dissolved in water and added to 55 °C-equilibrated nematode growth medium (NGM) agar at the concentration of 1 mM just before plates were poured. Levamisole-containing plates were seeded with OP50 *Escherichia coli*. Young adult worms were put on plates containing levamisole, animals were left overnight at 20°C, and paralyzed animals were then scored.

## Quantitative Real-Time PCR assays

Quantitative Real-Time PCR assays were performed as previously described (*Richard et al., 2013*). In *C. elegans*, total RNA was isolated from a synchronized population of worms using the RNeasy Kit (Qiagen) according to the manufacturer's instructions. In C2C12, myoblasts were cultured in 35 mm dishes and differentiated for 6 days before total RNA extraction using Trizol reagent (ThermoFisher). All samples were treated with DNase (Fermentas). First-strand cDNA was synthesized from 200 ng of total RNA using an oligo(dT) primer and a Moloney murine leukemia virus reverse transcriptase (Fermentas) at 42°C for 1 hr. Quantitative PCR was performed using LightCycler 480 SYBR Green I Master (Roche). A relative quantification model was used to evaluate the relative expression ratio of the target genes and RNA levels were normalized to three housekeeping genes (*cdc-42, pmp-3*, and *Y45F10D.4* for *C.elegans* RNAs; *HPRT1*, *CycloB* and glyceraldehyde 3-phosphate dehydrogenase (*Gapdh*),for mouse RNAs).

## Electrophysiology

Electrophysiological methods were performed as previously described by (*Richmond and Jorgensen, 1999*) and (*Richard et al., 2013*).

## Immunocytochemical staining

Worms were prepared by the freeze-crack method described previously (*Gendrel et al., 2009*). Methanol/acetone fixation was used for all staining conditions. The antibodies were used at the following dilutions: anti-UNC-38, 1:500 (*Gendrel et al., 2009*); anti-UNC-49, 1:500 (*Gally and Bessereau, 2003*); anti-VAChT/UNC-17, 1:1000 (*Gally et al., 2004*); the secondary antibodies were used at the following dilutions: Cy3-labeled goat anti-rabbit (1:1,000; Jackson ImmunoResearch Laboratories), A488-labeled goat anti-mouse (1:500; Molecular Probes), and A488-labeled goat anti-rat (1:1,000; Molecular Probes). Incubation conditions were the following: anti-UNC-38 and anti-UNC-49 overnight at 4°C, anti-UNC-17 for 1 hr at room temperature. Secondary antibodies were incubated together for 3 hr at room temperature.

## Cell culture and western blot

C2C12 cell lines were differentiated in myotubes and RT-PCR with primers specific for murine muscle was performed for cell identity confirmation. The cells were subjected to MycoAlert Mycoplasma detection kit from Lonza (LT07-318). C2C12 myoblasts were plated on Matrigel (Corning)-coated 35 mm dishes and grown in Hyclone DMEM (GE healthcare) containing 4.5 g/l glucose, supplemented with 15% FBS (PAA) and penicillin/streptomycin. Cells were maintained at 37°C in a saturated humidity atmosphere containing 5% $CO_2$. For the differentiation of C2C12 myoblasts, fetal bovine serum was replaced by 2% horse serum (Biowest) when myoblasts reached 70–80% confluence. Cells were differentiated for 6 days.

*Sh-Creld1* and control myoblast lines were obtained by selecting puromycin-resistant clones of C2C12 myoblasts, respectively transfected with shRNA clone set against mouse *Creld1* or scrambled control clone (GeneCopoeia).

## Western blot in C2C12 cells

C2C12 cells were lysed in RIPA buffer (Abcam) supplemented with Complete protease inhibitor cocktail (Roche) and passed through a needle (Qiagen) to disrupt DNA. Protein concentration was measured with a *DC* Protein Assay according to manufacturer instructions (Bio-Rad). 30 µg of sample were boiled in 40 µl sample buffer and were loaded on 4–12% pre-cast gel (Biorad) and fast-transferred into nitrocellulose membrane (Bio-Rad). Membrane was blocked with blocking buffer (5% Non Fat Dry Milk, 0.002% Tween in PBS). Incubation with primary antibodies was done overnight in blocking buffer at 4 C. After three washes with PBS- Tween 0.002%, membranes were incubated with secondary antibodies coupled with HRP for 1 hr at room temperature. Proteins were visualized using LumiLight reagents (Roche).

Western blot membranes were probed with rabbit antibody anti-Creld1 (1:500 dilution; Abcam), mouse commercial antibody anti-αAchR (1:500 dilution; BD Transduction Laboratories), mouse commercial anti-GAPDH (1:10000 dilution, Merck Millipore), and secondary HRP-conjugated goat anti-rabbit or goat anti-mouse antibody (1:3000 dilution; ThermoFisher Scientific).

## Surface labeling

For surface labeling of AChR, C2C12 cells were incubated with 200 nM αBT-biotin (Invitrogen) in PBS for 30 min on ice. Cells were intensively washed in PBS after αBT-biotin and lysed in RIPA lysis buffer (as described above). Streptavidin-agarose (Cell Signaling Technology) was used to recover biotinylated proteins. Quantitation of the Western blots was done by using the ChemiDoc MP Imaging System (Bio-Rad).

## Biochemistry in *C. elegans*

Protein extraction and Western blotting were performed as described (*Richard et al., 2013*); and coimmunoprecipitation of *C. elegans* extracts are described in (*Tu et al., 2015*). Further details are provided below.

*C. elegans* Protein extraction and Western Blotting

Mixed-stage populations were collected from OP50-seeded NGM plates. Worms were rinsed three times with NaCl and allowed to sediment on ice. Pellets were solubilized in Laemmli buffer with 2% (vol/vol) β-mercaptoethanol, boiled at 90°C for 10 min and centrifuged for 5 min at 15,700 × g. For glycosylation profiles, samples were treated with denaturation buffer for 10 min at 90°C, incubated with endoglycosidase H (EndoH; New England Biolabs) or PNGaseF (New England Biolabs) for 1 hr at 37°C, and subsequently treated with Laemmli buffer as described above. Membranes were imaged with a LAS4000 (GE Healthcare Life Sciences) luminescence detector and band intensity-quantified with ImageJ software.

Western blot membranes were probed with affinity-purified rabbit antibody anti-UNC-29 (1:1000 dilution; custom antibody), mouse commercial antibody anti-TUBULIN (1:1000 dilution; Sigma), mouse commercial anti-GFP (1:1000 dilution, Roche), and secondary HRP-conjugated goat anti-rabbit or goat anti-mouse antibody (1:3000 dilution; ThermoFisher Scientific) and revealed with Lumi-Light reagents (Roche).

## Co-immunoprecipitation of *C. elegans* extracts

A mixed stage population of worms (5 mL) was frozen at −80°C until use. For extraction, worm pellets were ground under liquid nitrogen and thawed in an equal volume of ice-cold worm lysis buffer (WLB: 50 mM HEPES, 50 mM KCl, 100 mM NaCl, 1 mM EDTA, 2% Triton X-100, 2 mM PMSF and one tablet of complete Protease inhibitor cocktail (Roche) in 10 mL) (*Tu et al., 2015*). The suspension was rotated gently for two hours at 4°C and centrifuged at 15,000 g for 20 min at 4°C to remove worm debris. The supernatant was diluted to a final concentration of 0.2% Triton X-100 with WLB. A prewashed 50 µL of anti-GFP-Trap-A beads (Chromotek, gta-100) was added and incubated overnight at 4°C with gentle rotation. The anti-GFP Trap A beads were collected by centrifugation at 1,000 g for 3 min at 4°C. The beads were washed three times with washing buffer (50 mM HEPES, pH 7.7, 50 mM NaCl) without Triton X-100. The immunoprecipitated proteins were eluted in Laemmli buffer with beta-mercaptoethanol, boiled for 10 min at 95°C, centrifuged for 10 min at 15,700 × g, separated by running 4–20% gradient Precise Protein Gel (ThermoScientifc, 25224) and analysed further by Western blotting. The primary anti-GFP mouse monoclonal antibody (Roche) and

anti-RFP mouse monoclonal antibody (ThermoFisher Scientific, MA5-15257) were used at a 1:1000 dilution. Horseradish peroxidase (HRP)-conjugated goat anti-mouse (K4000, Dako) was used as a secondary antibody at a 1:50 dilution.

### Identification of putative mixed disulphides using substrate-trapping mutants

Worm extracts were prepared as described in the protocol of co-immunoprecipitation with the exception that 200 mM NEM (N-Ethylmaleimide) were added to the WLB buffer during protein extraction of samples in order to assay under not reducing conditions. Moreover sample tested under not reducing conditions were eluted in Laemmli buffer without beta-mercaptoethanol.

### Microscopy and fluorescence quantification

Animals were mounted on 2% agarose pads, anaesthetized with 5 ml of M9 buffer containing 100 mM sodium azide and examined with either a Leica 5000B microscope equipped with a spinning disk CSU10 (Yokogawa) and a Coolsnap HQ2 camera, or a Nikon Eclipse Ti equipped with a spinning disk CSUX1-A1 (Yokogawa) and an Evolve EMCCD camera. Image analysis was performed with ImageJ.

For the quantitative analysis of $GABA_AR$ fluorescence, and L-AChR fluorescence in living worms, young adult animals were mounted on 2% agarose pads and immobilized using polybead microspheres (0.1 mm diameter, Polyscience, 00876–15) in M9 buffer. Quantification of synaptic $GABA_AR$ or L-AChR was achieved as described previously (*Tu et al., 2015*).

## Acknowledgements

We thank Hong Zhan for constructs, J Rand for the anti-UNC-17 antibodies, the *Caenorhabditis* Genetic Center (which is funded by NIH Office of Research Infrastructure Programs, P40 OD010440) and Dr. Shohei Mitani for strains. MD was supported by the AFM, MR was supported by the Association pour la Recherche contre le Cancer, CS was supported by a fellowship within the Postdoctoral Program of the German Academic Exchange Service, a Long-Term Postdoctoral Fellowship of the Human Frontier Science Program and the European Molecular Biology Organization, TB was supported by INSERM. This work was supported by the AFM-Téléthon (Grant MyoNeurALP), the Fondation pour la Recherche sur le Cerveau "Operation Espoir en tête 2013' and the Programme Avenir Lyon Saint-Etienne.

## Additional information

### Funding

| Funder | Grant reference number | Author |
|---|---|---|
| Association Française contre les Myopathies | Post-doctoral Fellowship 16451 | Manuela D'Alessandro |
| European Molecular Biology Organization | Long term Post-doctoral fellowship | Christian Stigloher |
| Institut National de la Santé et de la Recherche Médicale | Junior Grant | Thomas Boulin |
| Association Française contre les Myopathies | Myoneuralp | Jean-Louis Bessereau |
| Deutscher Akademischer Austauschdienst | Postdoctoral Program of the German Academic Exchange Service | Christian Stigloher |
| Fédération pour la Recherche sur le Cerveau | Operation Espoir en tête 2013 | JeanLouis Bessereau |
| Fondation ARC pour la Recherche sur le Cancer | 4th year PhD program 2011 | Magali Richard |
| Human Frontier Science Program | Long-Term Fellowship | Christian Stigloher |

The funders had no role in study design, data collection and interpretation, or the decision to submit the work for publication.

## Author contributions

Manuela D'Alessandro, Conceptualization, Data curation, Formal analysis, Funding acquisition, Investigation, Visualization, Methodology, Writing—original draft, Project administration; Magali Richard, Conceptualization, Data curation, Formal analysis, Funding acquisition, Investigation, Methodology, Project administration; Christian Stigloher, Conceptualization, Data curation, Formal analysis, Funding acquisition, Investigation, Methodology; Vincent Gache, Investigation, Methodology; Thomas Boulin, Conceptualization, Data curation, Formal analysis, Funding acquisition, Investigation, Methodology, Writing—review and editing; Janet E Richmond, Conceptualization, Formal analysis, Investigation, Methodology, Writing—review and editing; Jean-Louis Bessereau, Conceptualization, Resources, Data curation, Formal analysis, Supervision, Funding acquisition, Validation, Methodology, Writing—original draft, Project administration, Writing—review and editing

## Author ORCIDs

Manuela D'Alessandro (iD) http://orcid.org/0000-0002-2194-2836
Magali Richard (iD) http://orcid.org/0000-0003-3165-3218
Christian Stigloher (iD) https://orcid.org/0000-0001-6941-2669
Vincent Gache (iD) http://orcid.org/0000-0002-2928-791X
Thomas Boulin (iD) http://orcid.org/0000-0002-1734-1915
Jean-Louis Bessereau (iD) http://orcid.org/0000-0002-3088-7621

## Decision letter and Author response

Decision letter https://doi.org/10.7554/eLife.39649.016
Author response https://doi.org/10.7554/eLife.39649.017

# Additional files

## Supplementary files

• Source data 1. Source data related to *Figure 1*, *Figure 3*, *Figure 4*, *Figure 5* and *Figure 6*.
DOI: https://doi.org/10.7554/eLife.39649.013

• Transparent reporting form
DOI: https://doi.org/10.7554/eLife.39649.014

## Data availability

All data generated or analysed during this study are included in the manuscript and supporting files.

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
