## [Decision Letter]

Thank you for submitting your article "CRELD1 is an evolutionarily-conserved maturational enhancer of ionotropic acetylcholine" for consideration by *eLife*. Your article has been reviewed by Gary Westbrook as the Senior Editor, a Reviewing Editor, and two reviewers. The following individual involved in review of your submission has agreed to reveal his identity: Henry A. Lester (Reviewer #2). The reviewers have discussed the reviews with one another and the Reviewing Editor has drafted this decision to help you prepare a revised submission.

The reviewers and editor appreciated the identification and phenotypic characterization of a novel factor regulating the biogenesis of nicotinic acetylcholine receptors (AChRs) in *C. elegans* and also in mammals. The study is technically sound and there are no doubts about the quality of the results. The work shows an interesting new aspect of AChR biogenesis and the involvement of a new factor. Apart from some requested editorial changes (listed below), there were nevertheless three experimental aspects that the reviewers and editor suggest require further elaboration.

Essential revisions:

1) Exploring the regulation of *crld-1* with *unc-74*. Because *crld-1* mutants have a partial phenotype, there may be other players as well. An obvious candidate is UNC-74, which is a PDI as well, mutation of which eliminates all AChR biogenesis. This protein has been shown by the authors to be essential for expression of AChRs in a heterologous system (*Xenopus oocytes*) but has never been properly characterized in *C. elegans*. It is thus surprising that the authors do not analyze UNC-74 along with CRLD-1, as it could be that CRLD-1 shows a partial phenotype simply because the main player is UNC-74 (in whatever precise role in protein reduction/cysteine bridge formation is required during AChR biogenesis). Can the authors provide information about the expression and localization pattern of *unc-74*? And compare the effects of its removal on the effect of *crld-1* removal?

2) Figure 3A, C: The data presented do not exclude that AChR functional properties are altered and thus levamisole responses are different – while this may not have to do with a reduced expression level. The immunostaining suggests that, but from a single image, the reader cannot fully appreciate the differences between the genotypes. Can the authors exclude an effect that is both based on mislocalization *and* on reduced function that contributes to the electrophysiologically measured (reduced) currents? Are there ways to quantify expression levels via the fluorescent reporter tags?

3) Figure 5C: The PDI assay is rather indirect. It provides reasonably good indication that CRLD-1 is a PDI, but what is the nature of the bands in the non-reduced version of the gel in the C30A variant? If the hypothesis is true, it should contain levamisole resisitant subunits. This could possibly be tested with the approach indicated in Figure 4, coIP of UNC-29::RFP with CRLD-1::GFP and specific detection in Western blot. If the UNC-29::RFP protein is part of the cysteine-linked species the authors may be able to detect RFP in the high-molecular weight bands. Or, directly probe their gel with anti-UNC-29 or anti-UNC-38 antibodies.

– Discussion section: This section is lengthy and extends to areas not well touched by the content of the paper. Thus, some things appear overstated and could be toned down. E.g. the Discussion section speculates about how CRLD-1 helps AChR biogenesis, but not much direct data supporting functions as a chaperone are provided. Different functions of the two isoforms/splice variants of *crld-1* are based on their different localizations in the ER, but this is not based on more direct data. Also, notions of *Creld1* possibly affecting nicotine addiction should include a reference to Hosur et al. (2009).

_ Subsection “Disruption of the evolutionarily conserved gene *crld-1* confers partial resistance to the cholinergic agonist levamisole” 'these data demonstrate' or 'this data demonstrates'.

– Subsection “CRLD-1A and CRLD-1B are ubiquitously expressed and localize to the ER”: 'in most, if not all…'.

– Subsection “CRELD1 and CRELD2 are conserved proteins controlling different processes in the ER”: Notion of differential levamisole sensitivity of specific mutants. I did not get what the authors refer to: Taking out A isoform eliminates levamisole sensitivity and taking out B isoform leaves it completely intact. So, it is an all-or-none requirement for A isoform, in all alleles affecting this isoform, nothing in-between/differential.

– Figure 2. Scale bars are confusing and incomplete.

– Figure 3E. The increased nicotine response in *crld-1* looks significant and could be interesting.

– None of the three words, protein disulphide isomerase, are proper nouns. That the acronym PDI is capitalized does not require capitalization of the full phrase.

– "their very short C-terminal regions." Please provide exact length?

---

## [Author Response]

Essential revisions:1) Exploring the regulation of crld-1 with unc-74. Because crld-1 mutants have a partial phenotype, there may be other players as well. An obvious candidate is UNC-74, which is a PDI as well, mutation of which eliminates all AChR biogenesis. This protein has been shown by the authors to be essential for expression of AChRs in a heterologous system (Xenopus oocytes) but has never been properly characterized in C. elegans. It is thus surprising that the authors do not analyze UNC-74 along with CRLD-1, as it could be that CRLD-1 shows a partial phenotype simply because the main player is UNC-74 (in whatever precise role in protein reduction/cysteine bridge formation is required during AChR biogenesis). Can the authors provide information about the expression and localization pattern of unc-74? And compare the effects of its removal on the effect of crld-1 removal?

*unc-74* mutants were isolated by Jim Lewis based on complete insensitivity to levamisole (Lewis, 1980). *unc-74* encodes a putative PDI (E. Jorgensen, unpublished data; Boulin et al., 2008), which has been analyzed by the Jorgensen's lab for several years. They demonstrated that no L-AChR is left in *unc-74(0)*, and that UNC-74 is an ER protein that works cell autonomously to synthesize L-AChRs in muscle cells (EJ, personal communication). A manuscript containing this data is in preparation by the Jorgensen's lab. Because no L-AChR is left in the *unc-74(0)* mutant, the *unc-74(0); crld-1a(0)* double mutant will not bring additional information on CRLD-1A function.

*2) Figure 3A, C: The date presented do not exclude that AChR functional properties are altered and thus levamisole responses are different – while this may not have to do with a reduced expression level. The immunostaining suggests that, but from a single image, the reader cannot fully appreciate the differences between the genotypes. Can the authors exclude an effect that is both based on mislocalization* and *on reduced function that contributes to the electrophysiologically measured (reduced) currents? Are there ways to quantify expression levels via the fluorescent reporter tags?*

As requested, we quantified L-AChR and GABA_A_ receptors at synapses using knock-in strains expressing fluorescently-tagged UNC-29 and UNC-49, respectively. We found that the fluorescence intensity of L-AChRs present at the ventral nerve cord was decreased by 85% in *crld-1* mutant as compared to the wild type, while GABAR content was unchanged. Therefore, the reduction of the electrophysiologically measured currents in *crld-1* mutants can be explained solely by the reduction of L-AChRs present at NMJs and does not imply an additional impairment of L-AChR function.

These results are mentioned in the text (subsection “CRLD-1 is required for cell surface expression of L-AChRs”) and provided in Figure 3F-I.

3) Figure 5C: The PDI assay is rather indirect. It provides reasonably good indication that CRLD-1 is a PDI, but what is the nature of the bands in the non-reduced version of the gel in the C30A variant? If the hypothesis is true, it should contain levamisole resistant subunits. This could possibly be tested with the approach indicated in Figure 4, coIP of UNC-29::RFP with CRLD-1::GFP and specific detection in Western blot: If the UNC-29::RFP protein is part of the cysteine-linked species the authors may be able to detect RFP in the high-molecular weight bands. Or, directly probe their gel with anti-UNC-29 or anti-UNC-38 antibodies.

We perfectly agree with the reviewer’s predictions and we tried to do the proposed experiment. We used an *unc-29::RFP; crld-1::GFP* double knock-in strain to immunoprecipitate CRLD-1::GFP (WT and C30A) and then detect UNC-29::RFP. In one experiment out of three, we were indeed able to detect RFP in the high molecular weight bands under non-reducing conditions using the C30A variant (see Author response image 1). However, we failed in two other experiments and we think this is because we are at the detection limit of UNC-29-RFP in the IP product. Unfortunately, anti-UNC-38 antibodies do not work for western blot and anti-UNC-29 antibodies recognize nonspecific high-molecular weight bands. Because of the lack of reproducibility of this coIP experiment, we prefer to not include these data in the final version of the manuscript.

– Discussion section: This section is lengthy and extends to areas not well touched by the content of the paper. Thus, some things appear overstated and could be toned down. E.g. the Discussion section speculates about how CRLD-1 helps AChR biogenesis, but not much direct data supporting functions as a chaperone are provided. Different functions of the two isoforms/splice variants of crld-1 are based on their different localizations in the ER, but this is not based on more direct data. Also, notions of Creld1 possibly affecting nicotine addiction should include a reference to Hosur et al. (2009).

This part of the Discussion section has been significantly shortened and the reference was included in the text.

– Subsection “Disruption of the evolutionarily conserved gene crld-1 confers partial resistance to the cholinergic agonist levamisole”: 'these data demonstrate' or 'this data demonstrates'

This was corrected.

– Subsection “CRLD-1A and CRLD-1B are ubiquitously expressed and localize to the ER”: 'in most, if not all…'

This was corrected.

– Subsection “CRELD1 and CRELD2 are conserved proteins controlling different processes in the ER”: Notion of differential levamisole sensitivity of specific mutants. I did not get what the authors refer to: Taking out A isoform eliminates levamisole sensitivity and taking out B isoform leaves it completely intact. So, it is an all-or-none requirement for A isoform, in all alleles affecting this isoform, nothing in-between/differential.

"Differential sensitivity" was removed from the text

– Figure 2. Scale bars are confusing and incomplete.The legend of Figure 2 was clarified, accordingly.– Figure 3E. The increased nicotine response in crld-1 looks significant and could be interesting.

Although we agree that it might be interesting if nicotine response would be increased in *crld-1* mutants, this response is quite variable and there is no statistically significant difference between the *crld-1* mutant and the wild type. The main point we get from this set of data is that *crld-1* disruption does not cause a decrease of N-AChRs, as opposed to L-AChRs.

– None of the three words, protein disulphide isomerase, are proper nouns. That the acronym PDI is capitalized does not require capitalization of the full phrase.

This was corrected.

– "their very short C-terminal regions." Please provide exact length?

The exact length of the 2 isoform C-terminal regions is now provided in the text (subsection “CRLD-1A membrane topology”).